# Structure and dynamics of a mycobacterial type VII secretion system

Catalin M. Bunduc[1,2,3,4], Dirk Fahrenkamp[1,2,3], Jiri Wald[1,2,3], Roy Ummels[5], Wilbert Bitter[4,5], Edith N. G. Houben[4] & Thomas C. Marlovits[1,2,3 ✉]

*Mycobacterium tuberculosis* is the cause of one of the most important infectious diseases in humans, which leads to 1.4 million deaths every year[1]. Specialized protein transport systems—known as type VII secretion systems (T7SSs)—are central to the virulence of this pathogen, and are also crucial for nutrient and metabolite transport across the mycobacterial cell envelope[2,3]. Here we present the structure of an intact T7SS inner-membrane complex of *M. tuberculosis*. We show how the 2.32-MDa ESX-5 assembly, which contains 165 transmembrane helices, is restructured and stabilized as a trimer of dimers by the $MycP_5$ protease. A trimer of $MycP_5$ caps a central periplasmic dome-like chamber that is formed by three $EccB_5$ dimers, with the proteolytic sites of $MycP_5$ facing towards the cavity. This chamber suggests a central secretion and processing conduit. Complexes without $MycP_5$ show disruption of the $EccB_5$ periplasmic assembly and increased flexibility, which highlights the importance of $MycP_5$ for complex integrity. Beneath the $EccB_5$–$MycP_5$ chamber, dimers of the $EccC_5$ ATPase assemble into three bundles of four transmembrane helices each, which together seal the potential central secretion channel. Individual cytoplasmic $EccC_5$ domains adopt two distinctive conformations that probably reflect different secretion states. Our work suggests a previously undescribed mechanism of protein transport and provides a structural scaffold to aid in the development of drugs against this major human pathogen.

*Mycobacterium tuberculosis* encodes five homologous, but functionally distinct, T7SSs that are designated ESX-1 to ESX-5. These systems translocate a number of effector proteins across the unique and impermeable diderm cell envelope[2,3]. Because of their importance for mycobacterial physiology and virulence, T7SSs are considered to be promising targets for the development of drugs for the treatment or prevention of tuberculosis[4]. Although T7SSs have previously been shown to form hexameric complexes[5], high-resolution structural information exists only for part of the T7SS—a dimeric ESX-3 subcomplex from the nonpathogenic species *Mycobacterium smegmatis*[6,7]. Here we reconstituted the ESX-5 T7SS of *M. tuberculosis* H37Rv in *M. smegmatis* to obtain a structural view of the entire T7SS membrane complex from this human pathogen.

## Architecture and stoichiometry

The *M. tuberculosis* ESX-5 system showed robust expression in *M. smegmatis* and correct assembly of the membrane complex (Extended Data Fig. 1a, b). Purification of the *M. tuberculosis* ESX-5 membrane complex (using a C-terminal Strep tag on $EccC_5$ and mild solubilization conditions) resulted in copurification of the conserved $MycP_5$ protease (Extended Data Fig. 1c–e). To our knowledge, MycP (also known as mycosin) is absent in all previously reported T7SS structures[5–7]—although

MycP is known to be essential for T7SS function and complex stability[8]. The addition of nucleotides and $MgCl_2$ improved sample homogeneity, as judged by a more-distinct high molecular weight complex on native-PAGE, size exclusion chromatography and subsequent negative-stain electron microscopy analysis (Extended Data Fig. 1f–j).

Cryo-electron microscopy (cryo-EM) analysis of the *M. tuberculosis* ESX-5 complex purified in the presence of ADP–$AlF_3$ showed clear hexameric particles (Extended Data Figs. 2b, 3). We performed an ab initio reconstruction without symmetry enforcement that yielded an average resolution of approximately 4 Å (Extended Data Fig. 4), which improved to an overall resolution of approximately 3.5 Å after further data processing; this allowed us to build around 78% of the stable complex de novo (Supplementary Tables 1, 2). The intact machinery comprises $EccB_5$, $EccC_5$, $EccD_5$, $EccE_5$ and $MycP_5$ with a 6:6:12:6:3 stoichiometry (Fig. 1b, c, d, Supplementary Video 1) resulting in a 2.32-MDa complex that is anchored in the inner membrane through 165 transmembrane helices (TMHs) (Fig. 1e). The membrane assembly is best described as a trimer of dimers, in which each dimer comprises a single copy of $MycP_5$ and two protomers each of one copy of $EccB_5$, $EccC_5$, $EccE_5$ and two copies of $EccD_5$ (Fig. 1f). The overall fold and stoichiometry of a dimeric building block of *M. tuberculosis* ESX-5 is similar to that of a dimer of ESX-3 from *M. smegmatis*[6,7], albeit with notable differences on the periplasmic side and the angle between protomers. In the intact

[1]Centre for Structural Systems Biology, Hamburg, Germany. [2]Institute of Structural and Systems Biology, University Medical Centre Hamburg–Eppendorf, Hamburg, Germany. [3]Deutsches Elektron Synchrotron DESY, Hamburg, Germany. [4]Molecular Microbiology Section, Amsterdam Institute of Molecular and Life Sciences, Vrije Universiteit Amsterdam, Amsterdam, The Netherlands. [5]Department of Medical Microbiology and Infection Control, Amsterdam Infection and Immunity Institute, Amsterdam UMC, Amsterdam, The Netherlands. ✉e-mail: marlovits@marlovitslab.org

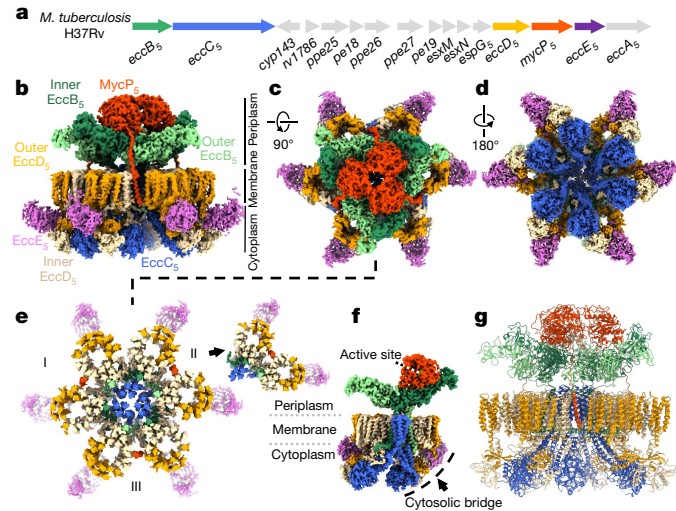

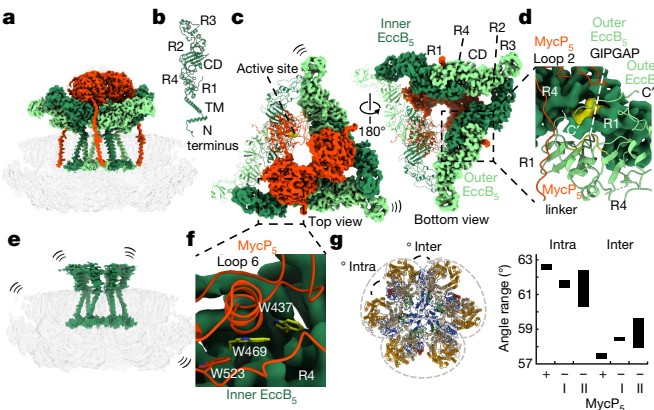

**Fig. 1 | Cryo-EM structure of the intact ESX-5 inner-membrane complex of *M. tuberculosis*. a**, Genetic organization of the *esx-5* locus of *M. tuberculosis* H37Rv, which was cloned and expressed in *M. smegmatis* MC²155. **b**–**e**, Cryo-EM density of the intact ESX-5 inner-membrane complex of *M. tuberculosis*, zoned and coloured for every individual component. Components are inner EccB₅ (dark green), outer EccB₅ (light green), EccC₅ (blue), inner EccD₅ (beige), outer EccD₅ (orange), EccE₅ (purple) and MycP₅ (red). The full complex is 28.5 nm in width and 20 nm in height, and has an absolute stoichiometry of 6:6:12:6:3 for EccB₅:EccC₅:EccD₅:EccE₅:MycP₅. **b**–**e**, Side (**b**), top (**c**) and bottom (**d**) views and a top cross-section (**e**) of the complex at the membrane level, highlighting the arrangement of the 165-TMH region. Inset, top cross-section of an extracted dimeric unit. **f**, Single dimer viewed from the centre of the intact complex, highlighting the central EccC₅ TMH bundle and the position of MycP₅ with its active site directed towards the inside of the periplasmic cavity. **g**, Ribbon model of the *M. tuberculosis* ESX-5 assembly.

ESX-5 complex, the angle between protomers at the membrane level differs by about 0.5°, from 59.7° between protomers of one dimer to 60.2° between protomers of adjacent dimers. However, at the cytosolic level these angles differ by more than 10° (from 65.3° to 54.7°) (Extended Data Fig. 5). By contrast, the individual ESX-3 dimer displayed an overall angle of 72° between protomers[6,7].

## Structural rearrangements of periplasmic domains

The periplasmic assembly of the ESX-5 membrane complex of *M. tuberculosis* is formed by three EccB₅ dimers and three MycP₅ proteases. The EccB₅ dimers assemble in a triangle, which forms a central cavity (Fig. 2c). Within an EccB₅ dimer, two slightly different conformations (that is, inner and outer) can be distinguished between monomers, depending on their position. EccB₅ dimerization is mediated mainly through the R1 and R4 repeat domains and is further stabilized by the EccB₅ C termini, which wrap around their interacting EccB₅ partner to form intermolecular hydrophobic contacts with its R1 domain (Fig. 2d). The GIPGAP motif—which is a highly conserved region in EccB homologues—is central to these interactions (Fig. 2d). Compared to the EccB₃ dimer from the ESX-3 subassembly[6,7], the three EccB₅ dimers are rotated by about 52° with respect to their corresponding EccC₅–EccD₅–EccE₅ membrane dimers, which indicates that large conformational rearrangements are required during maturation into the fully assembled hexamer (Extended Data Fig. 6d). To form the triangle-shaped assembly, the inner EccB₅ engages the outer EccB₅ of the adjacent dimer by packing its R3 domain against the α-helices α5 and α8 of domains R2 and R3, respectively, which results in an asymmetric EccB₅–tip arrangement (Fig. 2c). Consequently, domain R3 of the outer EccB₅ does not form any interactions at its tip extremity and thus displays higher flexibility, consistent with previous observations[9].

**Fig. 2 | MycP₅ drives EccB₅ hexamerization and stabilization of the membrane complex. a**, Transparent assembly of intact *M. tuberculosis* ESX-5, with EccB₅ and MycP₅ coloured as in Fig. 1. **b**, Complete structure of monomeric *M. tuberculosis* EccB₅, highlighting its overall fold and domains. **c**, Top and bottom view of the EccB₅–MycP₅ periplasmic assembly with one unit (EccB₅ dimer and MycP₅ monomer) as ribbon model, highlighting the active site of MycP₅ in yellow. CD, central domain; TM, TMH. **d**, EccB₅ dimerization site, highlighting the C-terminus of outer EccB₅ that is wrapped around the R1 and R4 domains of the adjacent inner EccB₅ monomer, the conserved GIPGAP motif of EccB₅ (in yellow) and the interactions of the EccB₅ dimer with loop 2 and the linker connection of MycP₅. **e**, Transparent map of *M. tuberculosis* ESX-5 without copurified MycP₅, with EccB₅ highlighted in dark green. The high flexibility of EccB₅ and the overall heterogeneity of the membrane complex in the absence of MycP₅ is indicated by curved lines. **f**, EccB₅–MycP₅ interaction surface, highlighting the three buried tryptophans. **g**, Angle variation range between protomers of the MycP₅-bound (+) and two unbound (−) states (I and II). Intra, between two protomers within a dimer: 62.7°, 62.4° and 62.4° (MycP₅-bound); 61.8°, 61.3° and 61.5° (unbound, I); 61.4°, 62.3° and 60.2° (unbound, II). Inter, between two protomers of adjacent dimers: 57.5°, 57.3° and 57.5° (MycP₅-bound); 58.5°, 58.3° and 58.3° (unbound, I); and 57.9°, 58.3° and 59.6° (unbound, II).

## Periplasmic MycP₅–EccB₅ assembly

Our periplasmic *M. tuberculosis* ESX-5 map shows three MycP₅ proteases that form a dome-like structure, which cap the periplasmic central cavity (Fig. 2a, c, Supplementary Video 2). Interactions between EccB₅ and MycP₅ are mediated mainly by the MycP₅ protease domain and a composite interface that is generated by the R4 domain and loop 6 (residues Thr424 to Ser435) of the inner EccB₅ (Fig. 2f). The MycP₅–EccB₅ interface covers a surface area of about 1,230 Å², which leads to the burial of three conserved tryptophan residues (Trp437 and Trp469 of EccB₅, and Trp523 of MycP₅) (Fig. 2f). Additionally, loop 2 of MycP₅ binds to the C terminus of the outer EccB₅, which explains why a deletion of this loop previously showed reduced ESX-5 secretion in *Mycobacterium marinum*[10] (Fig. 2d, Extended Data Fig. 6i). MycP₅–MycP₅ interactions are mediated mainly through loop 1 and the N-terminal extension (which run across the top of the MycP₅ protomers), and loop 3, which contacts the neighbouring protease domain from the side (Extended Data Fig. 7c, d). Loop 5 (residues Ala151 to Val271), which is cleaved during ESX-5 maturation[11], folds along the interface of two protease domains towards the central pore formed by the MycP₅ trimer (Extended Data Fig. 7a). Although we could not build a complete model of loop 5 (owing to its high flexibility), this loop appears to cap the central periplasmic pore (Extended Data Fig. 7a). Notably, as loop 5 is not present in all mycosins and is dispensable for ESX-5 secretion[10], a speculative role in gating remains to be identified. The active sites of the MycP₅ proteases face towards the central lumen of the cavity (Fig. 2c), which implies that potential substrates of this protease are translocated through—and processed within—this periplasmic chamber.

The dimer interface between the inner and outer EccB₅ is the largest in the periplasmic assembly, and covers a surface area of around

2,000 Å$^2$ and provides a solvation-free energy gain of $\Delta G = -23$ kcal mol$^{-1}$ per dimer (Fig. 2d). By contrast, the interfaces formed between EccB$_5$ dimers each bury a surface area of about 600 Å$^2$, with a cumulative energy gain of only $\Delta G = -18$ kcal mol$^{-1}$ upon trimerization. This could provide an explanation as to why dimeric ESX subcomplexes are more stable than their fully assembled counterparts[6,7]. The intermolecular EccB$_5$–MycP$_5$ interactions (which have a surface area of around 395 Å$^2$ and $\Delta G = -0.1$ kcal mol$^{-1}$) are even more modest, which provides a rationale for why interactions between MycP$_5$ and the membrane complex have so far remained unknown.

## MycP$_5$ stabilizes the entire membrane complex

To further investigate the effect of MycP$_5$ on the entire structure, we analysed MycP$_5$-free *M. tuberculosis* ESX-5 complexes from the same preparation (Fig. 2e, g). These assemblies contained the same EccB$_5$:EccC$_5$:EccD$_5$:EccE$_5$ stoichiometry as the fully assembled complexes (Extended Data Fig. 8). Following 3D reconstruction, we obtained two MycP$_5$-free *M. tuberculosis* ESX-5 maps that displayed resolution estimates of about 4.5 and about 6.7 Å (Extended Data Fig. 4). The differences were most notable on the periplasmic side, on which the six EccB$_5$ copies showed high flexibility and did not form a stable triangular scaffold in the absence of MycP$_5$ (Fig. 2e). This shows that MycP$_5$ enables the trimerization of the EccB$_5$ dimers in the periplasm. This result is highly interesting, because mycosins are subtilisin-like proteases without any additional domains apart from a TMH, some small loops and an N-terminal extension that wraps around the protein[12,13]. A more structural role for mycosin has previously been predicted[8] but we now understand the essential role of mycosins in T7SSs.

In contrast to the periplasmic domain, the cytosolic and membrane regions of the MycP$_5$-free maps were more similar to those of the MycP$_5$-containing particles (Extended Data Fig. 8). However, the MycP$_5$-free particles displayed an increased heterogeneity that affected the entire complex, which resulted in a slight waving of the membrane region and an increased angle variation between individual protomers (Fig. 2g, Extended Data Fig. 8, Supplementary Video 4). The protease domain and TMH of MycP$_5$ synergistically reinforce the membrane complex. Their interactions with periplasmic inner EccB$_5$ and membrane-embedded outer EccD$_5$ (from separate protomers within a dimer) better anchor the periplasmic assembly to the membrane, while also stabilizing the dimeric unit (Extended Data Figs. 8f, 9). Additionally, by driving the formation of the periplasmic assembly, MycP$_5$ stabilizes the entire complex by promoting cross-dimer MycP$_5$–MycP$_5$ and inner EccB$_5$–outer EccB$_5$ interactions (Extended Data Fig. 9). Our MycP$_5$-free *M. tuberculosis* ESX-5 maps are reminiscent of the hexameric, low-resolution negative-stain structure of ESX-5 from *Mycobacterium xenopi*, in which the periplasm was similarly disorganized in the absence of MycP$_5$[5].

## EccC$_5$ gates a potential secretion conduit

At the membrane level, six EccD$_5$-dimer barrels (each of which contains 22 TMHs) together form a circular raft with an inner cavity (Extended Data Fig. 10). Within this raft, inner EccD$_5$ monomers are situated closer to the centre, whereas outer EccD$_5$ monomers face towards the periphery of the membrane complex. The EccD$_5$ membrane barrels are structurally highly similar to the homologous EccD$_3$ barrel in the ESX-3 subassembly[6,7]. The inner surface of each EccD$_5$ barrel is decorated with densities that are attributable to stably bound lipids or detergent molecules, which suggests that, in their native membrane environment, these barrels are filled with membrane lipids (Extended Data Fig. 10e, f).

The TMH of each copy of EccB$_5$ is anchored within the confinement of the EccD$_5$ raft through hydrophobic interfaces that are provided by TMH6 and TMH11 of inner EccD$_5$ and stably bound lipids (Fig. 3a, Extended Data Fig. 6f). Notably, no substantial intermolecular

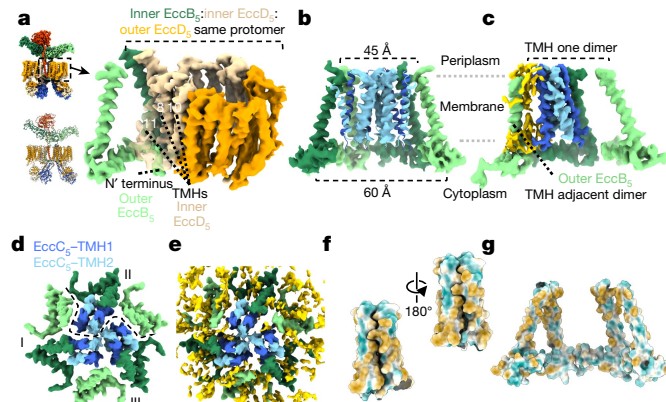

**Fig. 3 | A basket formed by the EccB$_5$ TMHs holds three four-TMH bundles of EccC$_5$. a**, Angled view from the outside of the complex, showing the TMH and N terminus of an outer EccB$_5$ interacting with a pocket formed by TMH8, TMH10 and TMH11 of inner EccD$_5$ from the adjacent barrel. **b**, Side cross-section through the EccB$_5$ basket that contains the EccC$_5$ TMH bundles. Light blue densities depict the three copies of EccC$_5$ TMH2 that form the central pyramid. Two TMHs of EccC$_5$ were removed for clarity. Sizes indicate the inner diameters of the EccB$_5$ basket. **c**, Side cross-section through an EccB$_5$ basket, showing that the EccC$_5$ TMH bundle does not interact with outer EccB$_5$ from its own dimer, but instead forms lipid-mediated interactions with the outer EccB$_5$ TMH of the adjacent dimer. Lipids are shown in gold. **d**, Top view of the central EccB$_5$ basket and the EccC$_5$ TMH bundles. Dashed line marks the TMHs that belong to one dimeric unit. **e**, As in **d**, highlighting the lipid-rich environment. In the central area that surrounds the EccC$_5$ pyramid, lipids are not clearly distinguishable (which suggests fluidity in this area). **f, g**, Surface model displaying the hydrophobicity of an EccC$_5$ TMH bundle (**f**) and the EccB$_5$ basket (**g**). Hydrophilic amino acids are shown in turquoise, and hydrophobic residues are shown in sepia.

interactions can be found between adjacent EccD$_5$ barrels. Instead, coupling between two neighbouring EccD$_5$ barrels is achieved by the N-terminal loop and α-helix of EccB$_5$ that run parallel to the cytoplasmic side of the inner membrane and engage in interactions with the TMHs of the neighbouring EccD$_5$ barrel in a clockwise manner (Fig. 3a, Extended Data Fig. 6g, h). Because the TMH of EccB$_5$ is slightly angled towards the centre of the complex, the architecture of the hexamer of EccB$_5$ TMHs is reminiscent of a basket, the inner diameter of which shrinks from around 60 Å to around 45 Å towards the periplasmic side (Fig. 3b).

EccC is the only component that is present in all T7SSs (including in related systems in Firmicutes[2]), and is therefore thought to be the central component in this nanomachinery. Each EccC protein has—in addition to two TMHs—four FtsK/SpoIIIE-like ATPase domains (also known as nucleotide-binding domains (NBDs)) that are known to be important for secretion[6,7,14]. We fully resolved the twelve EccC$_5$ TMHs in the intact *M. tuberculosis* ESX-5 complex; these form three four-TMH bundles, each of which belongs to the EccC$_5$ molecules of one dimer (Fig. 3b–d, Extended Data Fig. 11). These bundles are held together by hydrophobic interactions and effectively seal the central space of the membrane complex, which is enclosed by the EccB$_5$ basket (Fig. 3d, Supplementary Video 3). Two EccC$_5$ TMHs from each bundle contact the TMH of the inner EccB$_5$, which leaves the outer EccB$_5$ TMH unbound by EccC$_5$ (Fig. 3c, d). At the very centre of the complex, one TMH of each bundle contributes to the formation of a pyramidal assembly that aligns with the periplasmic chamber (Extended Data Fig. 11d).

The chamber within the EccB$_5$ basket appears to be filled with lipids. However, the density for these lipids is more ambiguous than that of the lipids in and around the EccD$_5$ barrels, which suggests that the lipids within this chamber are more fluid (Fig. 3e). Notably, the local resolution gradually increases when moving from the centre to the EccB$_5$ basket, where the resolution is highest (Extended Data Fig. 11c).

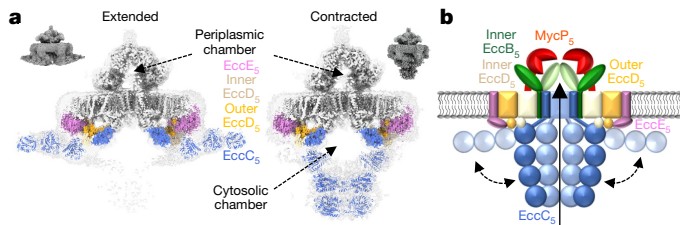

**Fig. 4 | EccC₅ adopts an extended and a contracted conformation. a**, Side cross-section of density maps, showing the extended and contracted conformation of EccC₅. The periplasmic and cytoplasmic chambers formed by EccB₅–MycP₅ and by EccC₅ upon closing are highlighted. Homology models of the three C-terminal NBDs of EccC₅ are fitted in the cytosolic densities. Cytosolic bridge components are coloured as in Fig. 1. **b**, Model of the intact T7SS inner-membrane complex, highlighting the two conformations of EccC₅.

This indicates that the EccC₅ TMH bundles display more flexibility, as compared to the rigid EccB₅ basket. The entrance to the putative EccC₅ pore widens on the cytoplasmic side, where the EccC₅ stalk domains expand radially (Extended Data Figs. 5, 11a). Together, our data suggest that the six EccD₅ barrels provide a stable scaffold for assembly of a secretion pore that is confined by the EccB₅ TMHs and gated through three EccC₅ TMH bundles. Secretion through the inner membrane complex would require rearrangement of the EccC₅ TMHs. Such a proposed central pore would extend into the periplasmic chamber that is formed by EccB₅ and MycP₅.

## Cytosolic EccC₅ adopts two conformations

At the cytoplasmic side of the complex, EccC₅ has a stalk helix that connects its second TMH to the first NBD (which is also known as the DUF domain). This NBD is bound to the cytosolic domains of inner and outer EccD₅, which—in turn—are bound to EccE₅ at the periphery, together forming a 'cytosolic bridge' (Fig. 1f).

The distal C-terminal part of EccC₅, which comprises a string of three NBDs (NBD1, NBD2 and NBD3), adopts two main conformations: we refer to these as extended and contracted (Fig. 4a). In the extended state, the C-terminal three NBDs of EccC₅ bend parallel to the membrane, and align with the cytosolic domains of inner EccD₅ and EccE₅ of the same protomer, and extend beyond the diameter of the membrane assembly. Although of considerably lower resolution, this density can confidently accommodate a homology model that consists of the three EccC₅ NBD domains (Fig. 4a, Extended Data Fig. 12). Further classification of the extended state reveals EccC₅ to be more heterogenous beyond NBD1, which suggests that NBD1 is more stably bound to components of its own protomer. Although we found only a relatively small number of particles in the contracted conformation, we solved the stable core of the membrane complex to sub-nanometre resolution (Extended Data Fig. 3). In the contracted state, the flexible arms of EccC₅ extend from the interface between the DUF domain of EccC₅ and the cytosolic domain of inner EccD₅ (Fig. 4a, Extended Data Fig. 12). We observed three separate disc-like structures that gradually constrict from the top to the bottom. This density shows a gap at the interface between NBD1 and NBD2. This would allow the previously postulated[14,15] binding of substrates to the linker 2 that connects NBD1 and NBD2, resulting in the displacement of this linker and the activation of NBD1. The highly dynamic cytoplasmic domains of the machinery

may provide the basis for substrate selection, recognition or transport across the membrane.

Our work provides a fully assembled structure of the ESX-5 inner-membrane complex of *M. tuberculosis*. As the membrane components of the five mycobacterial ESX systems show high sequence conservation, our results probably constitute a general structural blueprint for all of these T7SSs—including the virulence-related ESX-1 system (Supplementary Figs. 2, 3). Furthermore, our structure will serve as a platform for the identification of interactions that—if perturbed by small molecules—would aid in the treatment of tuberculosis.

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

## Methods

No statistical methods were used to predetermine sample size. The experiments were not randomized, and investigators were not blinded to allocation during experiments and outcome assessment.

### Molecular biology

*Escherichia coli* Dh5α was grown at 37 °C and 200 rpm in LB medium supplemented with 30 µg ml$^{-1}$ streptomycin. Cloning was performed in *E. coli* Dh5α using IProof DNA polymerase from BioRad and restriction enzymes from New England Biolabs. A list of the primers used for amplification is available in Supplementary Table 3.

The plasmid expressing *M. tuberculosis* ESX-5 was built as follows: the backbone of the previously described pMV ESX-5$_{mxen}$ plasmid[5] was modified to encode the unique restriction sites DraI and PacI upstream and SpeI and NdeI downstream of the TwinStrep tag sequence. The *rv1782–rv1783* (*eccB$_5$–eccC$_5$*) region, including about 380 bp upstream of *eccB$_5$*, of *M. tuberculosis* H37Rv was amplified while adding DraI and PacI restriction sites at the 5′ and 3′ ends, respectively (primers 1 and 2), and cloned into the modified plasmid upstream of the TwinStrep tag sequence, resulting in plasmid intermediate 1. The *M. tuberculosis* H37Rv region spanning *rv1791–rv1798* (*pe19–eccA$_5$*) was amplified while adding SpeI and NdeI unique restriction sites (primers 3 and 4) and cloned downstream of the TwinStrep tag sequence into the intermediate 1, resulting in plasmid intermediate 2. Plasmid intermediate 2 was digested with SpeI and SnaBI, removing the region *rv1791–rv1794* (*pe19–espG$_5$*), and the region encompassing *rv1785–rv1794* (*cyp143–espG$_5$*) was amplified as two individual PCR products (primers 5 and 6 and primers 7 and 8). The restricted backbone and PCR products were InFusion (Takara Bio)-ligated, resulting in the final pMV-ESX-5$_{mtb}$ containing the entire *rv1782–rv1798* (*eccB$_5$–eccA$_5$*) locus.

### Isolation of mycobacterial cell envelopes

*Mycobacterium smegmatis* MC$^2$155 expressing *M. tuberculosis* ESX-5 was grown at 37 °C and 90 rpm in LB medium supplemented with 0.05% Tween 80 and 30 µg ml$^{-1}$ streptomycin. Cultures were grown to an optical density (OD) at 600 nm of about 1.5, spun down for 15 min at 12,000$g$ in a JLA-8.1000 rotor and subsequently washed in PBS. After culture collecting, all subsequent steps were performed at 4 °C. Washed cell pellets were resuspended in buffer A (50 mM Tris-HCl pH 8, 300 mM NaCl and 10% glycerol) at a concentration of about 50 OD ml$^{-1}$ and lysed by passing two times through a high-pressure homogenizer (Stansted) using a pressure of 0.83 kbar. Unbroken cells were pelleted at 5,000$g$ for 5 min and supernatants were transferred to ultracentrifugation tubes. Cell envelopes were separated from the soluble fraction by ultracentrifugation at 150,000$g$ for 1.5 h. Following ultracentrifugation, supernatants were discarded, pellets were washed once with buffer A, resuspended in buffer A at a concentration of 750–1,000 OD ml$^{-1}$, snap-frozen in liquid nitrogen and stored at −80 °C until further use. The protein concentration of the cell envelope fraction was measured by BCA assay (Pierce).

### Purification of the *M. tuberculosis* ESX-5 membrane complex

All steps were performed at 4 °C. The *M. tuberculosis* ESX-5 was purified as follows: cell envelope fractions were diluted to 3 mg ml$^{-1}$ in buffer B (50 mM Tris-HCl pH 8, 300 mM NaCl and 5% glycerol), supplemented with 0.25% DDM, 3 mM ADP–AlF$_3$ and 6 mM MgCl$_2$. Following solubilization, the cell envelope mixture was spun down at 100,000$g$ for 20 min, supernatants were collected and incubated with StrepTactin resin (IBA). Beads were subsequently washed with buffer B supplemented with 0.03% DDM, 1 mM ADP–AlF$_3$ and 2 mM MgCl$_2$. Bound protein was eluted from the resin with buffer B supplemented with 0.03% DDM, 3 mM ADP–AlF$_3$, 6 mM MgCl$_2$ and 10 mM desthiobiotin. The protein concentration of the eluate was measured by Bradford assay and amphipol A8-35 was added in an amphipol:protein ratio of 5:1. After a 1-h incubation, the amphipol-containing eluate was incubated overnight (around 12–16 h) with BioBeads in a BioBeads:detergent ratio of 20:1. Subsequently, BioBeads were removed using gravity flow chromatography columns and the sample was concentrated using Amicon Ultra 0.5-ml 100-kDa spin concentrators. The concentrated sample was further purified through size exclusion chromatography (SEC), using a Superose 6 Increase column running in buffer C (20 mM Tris-HCl pH 8, 200 mM NaCl) supplemented with 1 mM ADP–AlF$_3$ and 2 mM MgCl$_2$. Size exclusion chromatography fractions were analysed by blue-native polyacrylamide gel electrophoresis (BN-PAGE) and negative-stain electron microscopy, after which the appropriate fractions were concentrated for cryo-EM using Amicon Ultra 0.5-ml 100-kDa spin concentrators. The initial Arctica sample was purified similarly, with the addition of 5% glycerol in the SEC purification step and the omission of ADP–AlF$_3$ and MgCl$_2$ from the purification protocol.

### BN-PAGE

Samples consisting of either solubilized membranes or purified membrane complexes were mixed with 5% G-250 sample additive (Invitrogen), to a final concentration of about 0.2%, and run on 3–12% NativePage Bis-Tris Protein Gels (Invitrogen) according to manufacturer specifications. Gels were either stained with Coomasie R-250 or transferred to PVDF membranes and stained with appropriate antibodies (Supplementary Fig. 1). Antisera against EccB$_5$, used in Extended Data Fig. 1b, was raised against the synthetic peptide CLPMDMSPAELVVPK and has previously been described[16]. Polyclonal rabbit antisera against the peptide was raised in rabbits by Innovagen using Stimune (Prionix) as adjuvants. The antibody was used as a 1:5,000 dilution. Blots were visualized on a ChemoStar TouchMotionCor2 using ChemoStarTS.

### Negative-stain electron microscopy

Carbon-coated copper grids were glow-discharged for 30 s at 25 mA using a GloQube Plus Glow Discharge System (Electron Microscopy Sciences). Four microlitres of diluted sample was applied to the grids and incubated for 30 s. The sample was blotted off from the side and the grid was washed briefly with 4 µl of staining solution (2% uranyl acetate) and then stained with 4 µl of the staining solution for 30 s. The stain was blotted off from the side and grids were air-dried. Grids were imaged using a Thermo Fisher Scientific Talos L120C TEM equipped with a 4K Ceta CEMOS camera using TIA 4.1.5.

### Cryo-EM sample preparation

For the main datasets, purified sample was applied to Quantifoil R2/2, 200 mesh, copper grids floated with an additional approximately 1.1-nm layer of amorphous carbon. Four microlitres of sample was applied onto glow-discharged grids (30 s at 25 mA) and allowed to disperse for 60 s at 4 °C and 100% humidity. Grids were blotted for 4–6 s with a blot force of −5 and plunge-frozen in a liquid propane–ethane mixture, using a Thermo Fisher Scientific Vitrobot Mark V. For the initial Arctica dataset, all steps were similar, with the exception of the additional layer of amorphous carbon.

### Cryo-EM data acquisition

The initial cryo-EM dataset was collected on a 200-kV FEI Talos Arctica electron microscope equipped with a Falcon III direct electron detector running in counting mode and using Thermo Fisher Scientific EPU 1.11. A total of 853 movies were recorded with a nominal magnification of 150,000×, corresponding to a pixel size of 0.96 at the specimen level. Movies were recorded with a total dose of 40.28 electrons per A$^2$, fractionated in 38 frames over a 40-s exposure time and with a nominal defocus range of 1–2.5 µm.

The two high-resolution datasets were recorded using Thermo Fisher Scientific EPU 2.4 software on a 300-kV Titan Krios TEM, equipped with a Gatan K3 direct electron detector running in counting mode and a Gatan Bioquantum energy filter (slit size 10 eV). We recorded 7,984

and 9,389 movies in counting mode in the two separate sessions with a nominal magnification of 81,000×, corresponding to a pixel size of 1.1 Å at the specimen level. Movies were recorded with a total dose of 59.5 electrons per $A^2$, fractionated in 50 frames over a 3-s exposure time and with a nominal defocus range of 1–3 μm.

## Cryo-EM data processing

Single-particle analysis was performed using Relion3.1[17], unless stated otherwise. For the initial Arctica dataset, movies were motion-corrected using MotionCor2[18] and dose-weighted, and the contrast-transfer function (CTF) was estimated using CTFFIND4[19]. Automated particle-picking was performed using Cryolo[20] and the pretrained Janni model. Following particle extraction and several rounds of 2D classification to remove obvious artefacts, an initial de novo model was generated. The dataset was further cleaned using 3D classification and the best class was subsequently used for reference-based particle-picking. Following 2D and 3D classification (and 3D refinement in C1), the map displayed an apparent threefold symmetry and was further refined in C3. This final map displayed an estimated 13.5 Å resolution.

In the first Krios dataset, movies were motion-corrected using Motion-Cor2, dose-weighted and the CTF was estimated using CTFFIND4. Automated particle-picking was performed using Cryolo with the pretrained Janni model and a low threshold. Particles were extracted and binned 4× and several rounds of 2D classification were performed followed by 3D classification with the 30 Å-filtered Arctica model as a template. The resulting particles were re-extracted without binning, CTF-corrected and polished and refined in C1, giving a map with an estimated overall resolution of 4.5 Å. For the cytosolic region, particles were recentred on the cytosolic region, re-extracted, CTF-corrected, polished and 3D-refined. Following refinement, the density accounting for individual cytosolic dimers was subtracted, resulting in a particle stack that was three times larger. Cytosolic dimers were first refined with a mask encompassing both cytosolic bridges. Subsequently, these were focus-refined using a soft mask around one of the cytosolic bridges. This map was refined using the default Relion value '--tau2fudge 2' but also '--tau2fudge 4', which increased the overall connectivity of the lower cytosolic area. The final map for the cytosolic bridge showed an estimated resolution of 3.3 Å and was sharpened using either Relion postprocessing or DeepEMhancer[21]. DeepEMhancer further helped to improve the observed anisotropy, overall map connectivity.

The second Krios dataset was processed similarly, with some exceptions. Following 3D classification of the binned data against the 4.5 Å Krios map filtered to 30 Å, the two maps with and without MycP₅ were processed separately. The MycP₅-unbound map displayed increased heterogeneity and—following unbinned re-extraction and refinement—the particles were 3D-classified without alignment, resulting in two obvious classes of 4.5 Å and 6.7 Å resolution. Model free density modification in Phenix.Resolve_Cryo_EM[22] further improved the resolution to 4.3 Å and 5.8 Å, respectively. By contrast, a similar 3D classification for the MycP₅-bound map did not further classify into structurally distinct classes. Following unbinned re-extraction and refinement, the MycP₅-bound map showed an overall resolution of 4 Å, which was further improved to 3.8 Å after C3 refinement. Model free density modification in Phenix.Resolve_Cryo_EM further improved the resolution of the entire C1 map to 3.8 Å and of the C3 refined map to 3.56 Å. For the periplasmic map, the centre of mass for that region was determined using Chimera[23] and the particles were recentred, extracted, 3D-refined and polished to obtain the periplasmic map at an estimated 3.8 Å resolution in C1. Following 3D classification without alignment and further refinement in C3, the estimated resolution of the periplasmic map improved to 3.5 Å. To separate the two states of EccC₅, particles were recentred on the lower cytosolic region, at the level of the DUF domain, polished and 3D-refined. This was followed by a masked 3D classification in which the mask contained NBD1 and NBD2 of EccC₅ in the extended conformation. The two main classes

were further 3D-refined unmasked and subsequently masked, leading to a map of about 4.27 Å for the extended conformation and 7.6 Å for the contracted conformation.

## Model building and refinement

Model building started by generating homology models for MycP₅, EccB₅, EccC₅ and EccD₅ with Phyre2[24]. For MycP₅, Protein Data Bank (PDB) entry 4J94[12] served as a structural template, and PDB entries 4KK7[25], 4NH0[14] and 6SGW[6], and 6SGZ[6] served as reference models for EccB₅, EccC₅ and EccD₅, respectively. To obtain atomic models of the periplasmic part (MycP₅–EccB₅) of the *M. tuberculosis* ESX-5 complex, homology models of MycP₅ and EccB₅ were rigid-body-fitted into a C1 symmetry, focus-refined periplasmic *M. tuberculosis* ESX-5 map (Electron Microscopy Data Bank (EMDB) code EMD-12518) using the fit-in-map tool in ChimeraX (v.1.0)[26]. Model building, extension and interactive refinement was performed with ISOLDE (v.1.0.1)[27], a molecular-dynamics-guided structure refinement tool within ChimeraX (v.1.0). The resulting coordinate file (PDB 7NPS) was further refined with Phenix.real_space_refine (v.1.18.2-3874)[28] using reference model restraints, strict rotamer matching and disabled grid search. Model validation was carried out using the MolProbity web server[29] and EMRinger[30] within the Phenix software package. Models for the membrane-embedded region (MycP₅–EccB₅–EccC₅–EccD₅) and cytoplasmic bridge (cytosolic domains of EccC₅–EccD₅) (PDB 7NPT) were built in the same way, using a reconstruction of the full *M. tuberculosis* ESX-5 complex (EMDB EMD-12517) and a focus-refined map of the cytoplasmic domains (EMDB EMD-12520) sharpened with DeepEMhancer, respectively. Finally, a composite model was assembled by fusing the periplasmic assembly and six copies of the cytosolic bridge to the membrane-embedded region model. This composite model was then refined against the full *M. tuberculosis* ESX-5 complex map with C1 symmetry (PDB 7NP7 and EMDB EMD-12514) and C3 symmetry (PDB 7NPR and EMDB EMD-12517).

Modelling into MycP₅-free maps was performed with ISOLDE using the composite ESX-5 model, in which MycP₅ and the periplasmic domain of EccB₅ (residues 84–507) had been deleted. Adaptive distance restraints as well as torsion restraints were applied to all atoms to restrain short-range conformational changes but allow for long-range conformational movements. ISOLDE simulations for dynamic fitting of the coordinate file into EMD-12521 and EMD-12522 were performed (about 10 min, 10 K) after which the models showed satisfying fits to the new maps without further manual intervention. MycP₅-free models were further refined against the maps using Phenix.real_space_refine (v.1.18.2-3874) as stated.

Visualization of atomic coordinates and map volumes was performed with ChimeraX (v.1.1) and PyMOL v.2.40[31]. Buried surface areas between subunits were calculated by PISA[32].

## Reporting Summary

Further information on research design is available in the Nature Research Reporting Summary linked to this article.

## Data availability

Cryo-EM maps have been deposited in the EMDB under accession codes EMD-12514 (full complex in C1), EMD-12517 (full complex in C3), EMD-12518 (periplasmic map in C1), EMD-12519 (periplasmic map in C3), EMD-12520 (cytosolic bridge), EMD-12521 (MycP₅-free map 1), EMD-12522 (MycP₅-free map 2), EMD-12523 (EccC₅, extended state) and EMD-12525 (EccC₅, contracted state). The composite model settled in the C1 and C3 full maps, periplasm in C1, cytosolic bridge, MycP₅-free map 1 and MycP₅-free map 2 have been deposited in the PDB under accession codes 7NP7, 7NPR, 7NPS, 7NPT, 7NPU and 7NPV, respectively. All other data are available from the corresponding author upon reasonable request.

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

**Acknowledgements** We thank all members of the laboratories of T.C.M., E.N.G.H. and W.B. for their support of this project; W. Lugmayr for scientific IT support; L. Ciccarelli for initial cryo-EM analysis of the *M. tuberculosis* ESX-5 complex; and T. Croll for his support with ISOLDE. High-performance computing was possible through access to the HPC at DESY/Hamburg. Cryo-EM data collection was performed at the Cryo-EM Facility at CSSB. This project was supported by funds available to T.C.M. through the Behörde für Wissenschaft, Forschung und Gleichstellung of the city of Hamburg at the Institute of Structural and Systems Biology at the University Medical Center Hamburg–Eppendorf (UKE). The laboratory of T.C.M. is supported by DESY (German Electron Synchrotron Center). The cryo-EM facility is supported by the University of Hamburg, the University Medical Center Hamburg–Eppendorf and DFG grant numbers INST152/772-1, 152/774-1, 152/775-1, 152/776-1 and 152/777-1 FUGG. This work received funding by a VIDI grant (864.12.006; to C.M.B. and E.N.G.H.) from the Netherlands Organization of Scientific Research. This project has received funding from the European Union's Horizon 2020 research and innovation programme under the Marie Sklodowska-Curie grant agreement no. 101030373 (to C.M.B.).

**Author contributions** C.M.B., D.F., J.W., E.N.G.H., W.B. and T.C.M. designed experiments. R.U. and C.M.B. generated constructs. C.M.B. purified complexes and performed biochemical assays. J.W. vitrified samples and collected cryo-EM images. C.M.B. and J.W. collected negative-stain images. D.F. built the atomic model. C.M.B., D.F., E.N.G.H., W.B. and T.C.M. interpreted data. C.M.B. and T.C.M. processed cryo-EM data. C.M.B., D.F., E.N.G.H., W.B. and T.C.M. wrote and revised the paper. All authors read, corrected and approved the manuscript. E.N.G.H., W.B. and T.C.M. supervised the project.

**Funding** Open access funding provided by Deutsches Elektronen-Synchrotron (DESY).

**Competing interests** The authors declare no competing interests.

**Additional information**
**Correspondence and requests for materials** should be addressed to T.C.M.

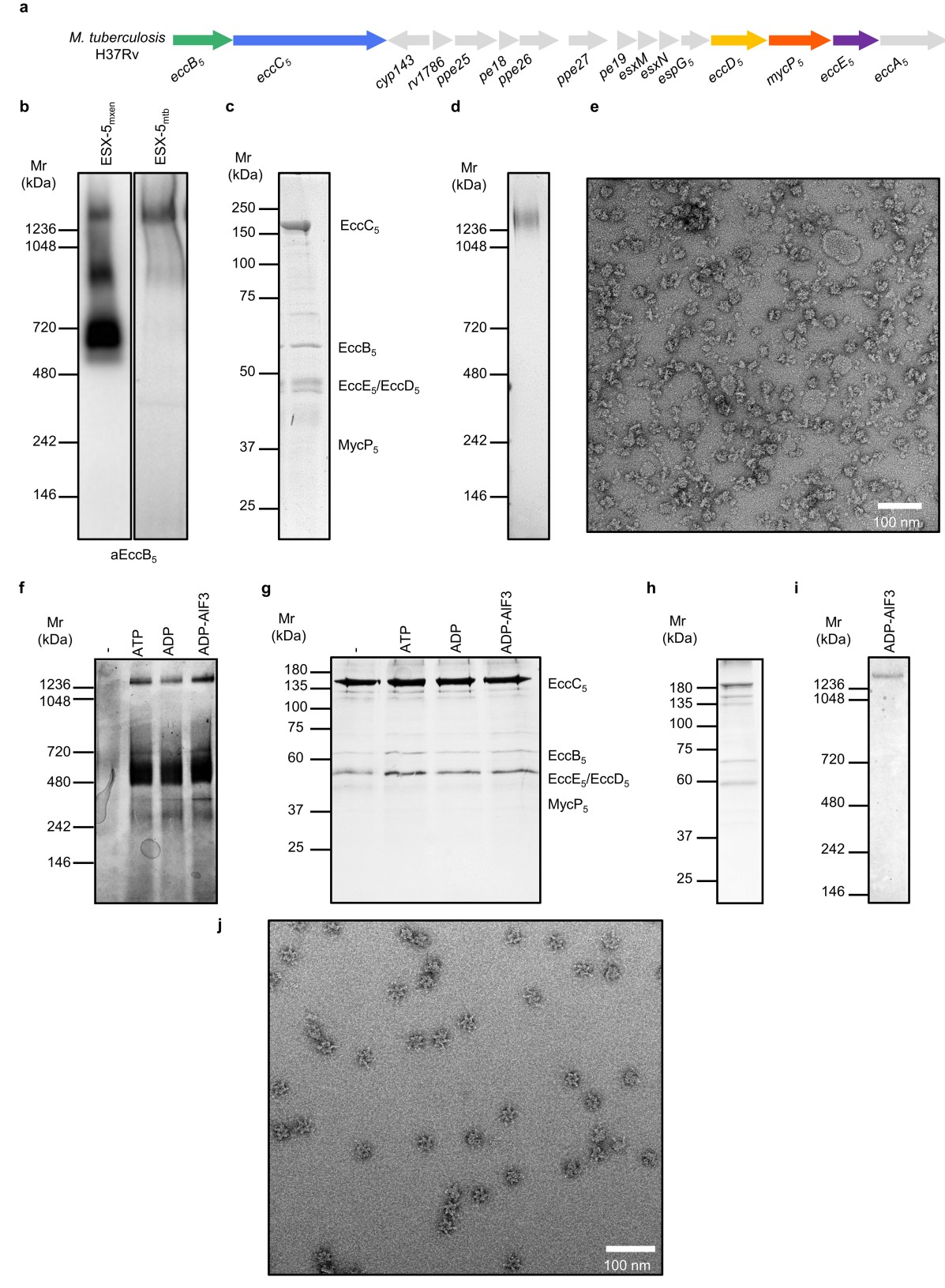

**Extended Data Fig. 1 |** See next page for caption.

**Extended Data Fig. 1 | Purification of the *M. tuberculosis* ESX-5 membrane complex. a**, Genetic organization of the *esx-5* locus of *M. tuberculosis* H37Rv, which has been cloned and expressed in *M. smegmatis* MC²155. **b**, BN-PAGE and western blot analysis using an anti-EccB$_5$ antibody of DDM-solubilized membranes from *M. smegmatis* MC²155 expressing *M. xenopi* or *M. tuberculosis* ESX-5 (ESX-5$_{mxen}$ or ESX-5$_{mtb}$, respectively). Experiment was reproduced three times. **c**, **d**, Coomassie-stained SDS–PAGE (**c**) and BN-PAGE (**d**) of Strep- and SEC-purified ESX-5$_{mtb}$ membrane complexes. **e**, Negative-stain electron microscopy analysis of ESX-5$_{mtb}$ membrane complexes shown in **c** and **d**. Experiments in **c**–**e** were replicated three times. **f**, BN-PAGE and Coomassie staining of Strep-purified ESX-5$_{mtb}$ complexes without nucleotides (−) or in the presence of nucleotides ATP, ADP or the transition-state analogue ADP–AlF$_3$. Upon purification, in the presence of either nucleotide, the higher-molecular-weight species of the membrane complex becomes more prominent. **g**, SDS–PAGE and Coomassie staining of the same samples as in **f**, showing a similar SDS–PAGE protein pattern between the four conditions. Experiment in **f**, **g** was performed three times. **h**, **i**, Coomassie-stained SDS–PAGE (**h**) and BN-PAGE (**i**) of Strep- and SEC-purified ESX-5$_{mtb}$ membrane complexes in the presence of ADP–AlF$_3$. **j**, Negative-stain electron microscopy of the same sample as in **h**, **i**, showing improved sample homogeneity as compared to purifications in the absence of nucleotides, as shown in **e**. Experiment shown in **h**–**j** was performed twice.

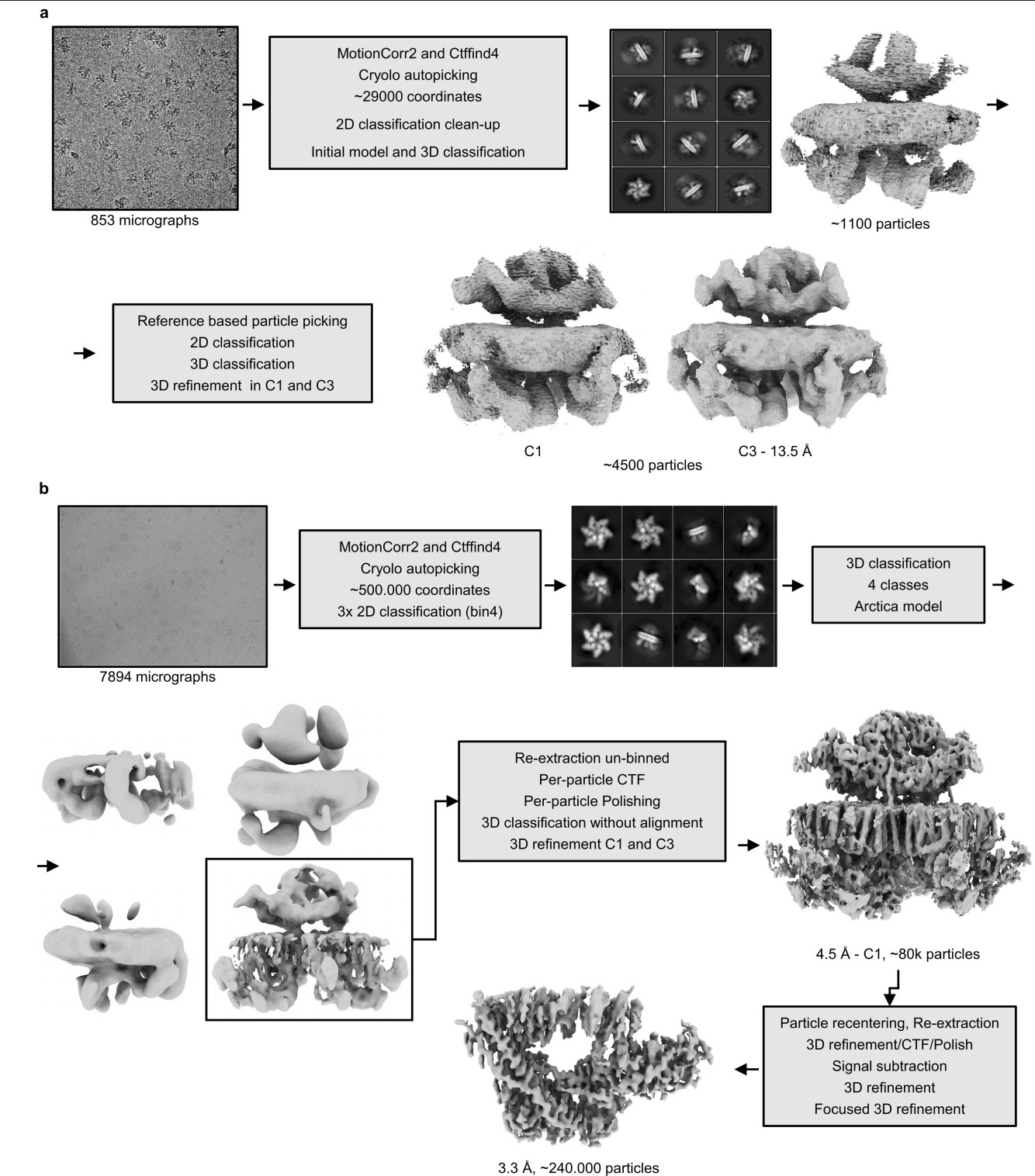

**a**

853 micrographs

MotionCorr2 and Ctffind4
Cryolo autopicking
~29000 coordinates

2D classification clean-up

Initial model and 3D classification

~1100 particles

Reference based particle picking
2D classification
3D classification
3D refinement in C1 and C3

C1

C3 - 13.5 Å

~4500 particles

**b**

7894 micrographs

MotionCorr2 and Ctffind4
Cryolo autopicking
~500.000 coordinates
3x 2D classification (bin4)

3D classification
4 classes
Arctica model

Re-extraction un-binned
Per-particle CTF
Per-particle Polishing
3D classification without alignment
3D refinement C1 and C3

4.5 Å - C1, ~80k particles

Particle recentering, Re-extraction
3D refinement/CTF/Polish
Signal subtraction
3D refinement
Focused 3D refinement

3.3 Å, ~240.000 particles

**Extended Data Fig. 2 | Cryo-EM data collection and single-particle reconstruction procedure. a, b**, This figure relates to the initial Talos-Arctica-collected dataset (**a**) and the first higher-resolution Titan-Krios-collected dataset (**b**).

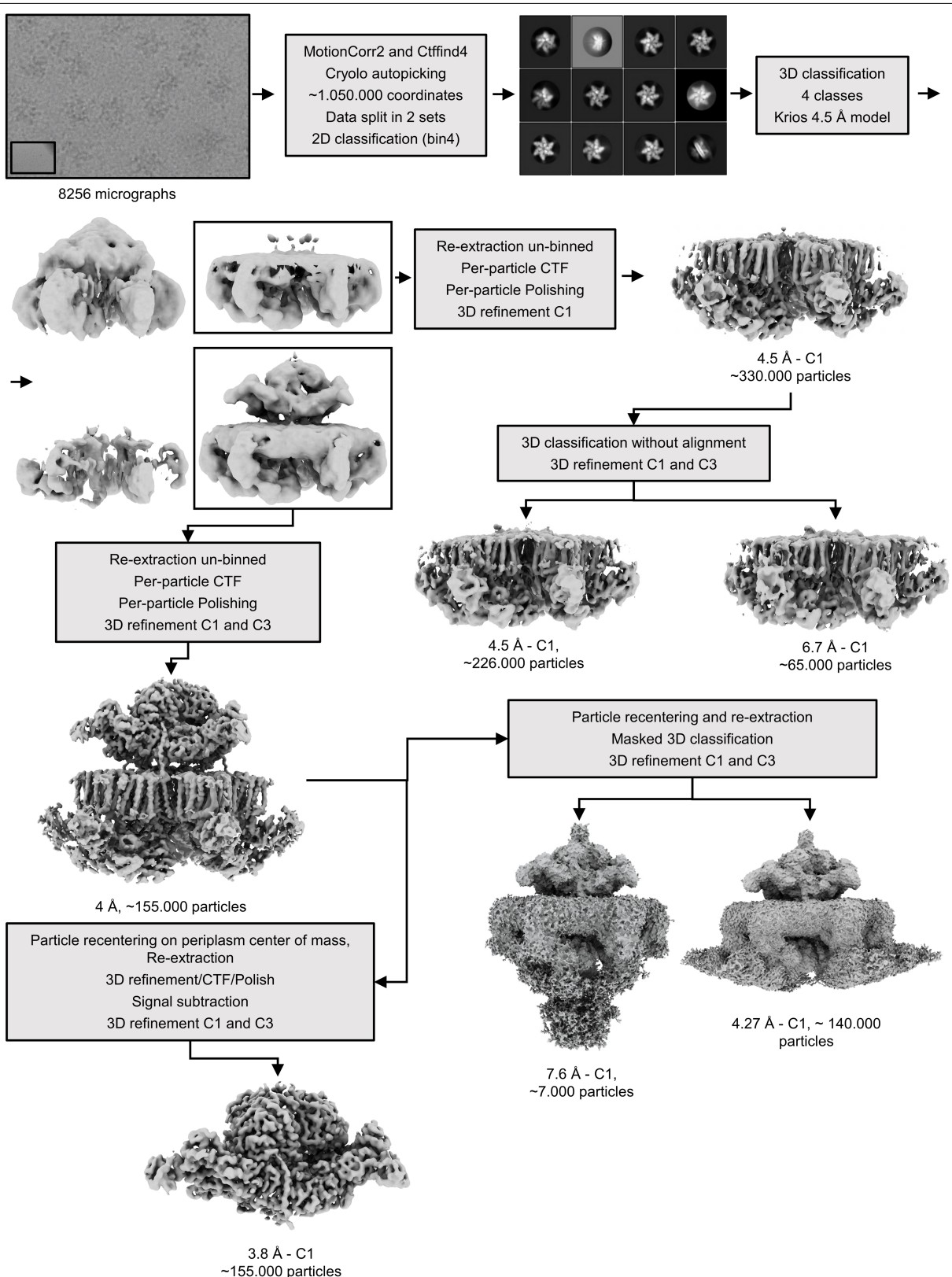

**Extended Data Fig. 3 | Cryo-EM data collection and single-particle reconstruction procedure.** This figure relates to the second high-resolution Titan Krios collected dataset.

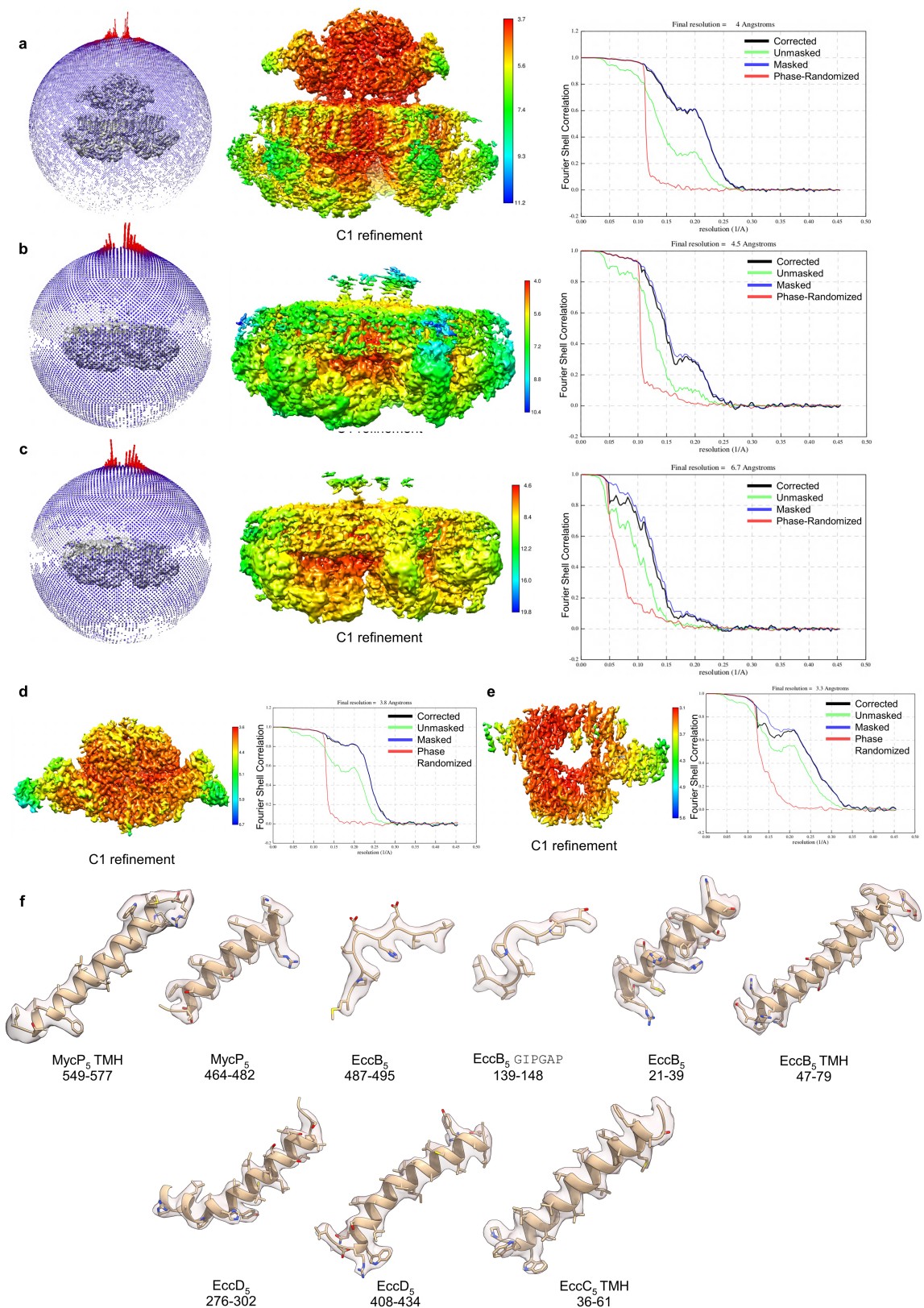

**Extended Data Fig. 4 | Single-particle reconstructions of the ESX-5$_{mtb}$ membrane complex. a–c**, Angular distribution plots, local-resolution estimations and Fourier shell correlation (FSC) plots of the C1 reconstruction of the entire MycP$_5$-bound ESX-5$_{mtb}$ membrane complex (**a**), and C1 reconstructions of the two heterogeneous MycP$_5$-unbound ESX-5$_{mtb}$ membrane complexes (**b**, **c**). **d**, **e**, Local-resolution estimation and FSC plot for the C1-refined periplasmic map (**d**) and the map of the cytosolic bridge (**e**). **f**, Examples of cryo-EM densities and corresponding models.

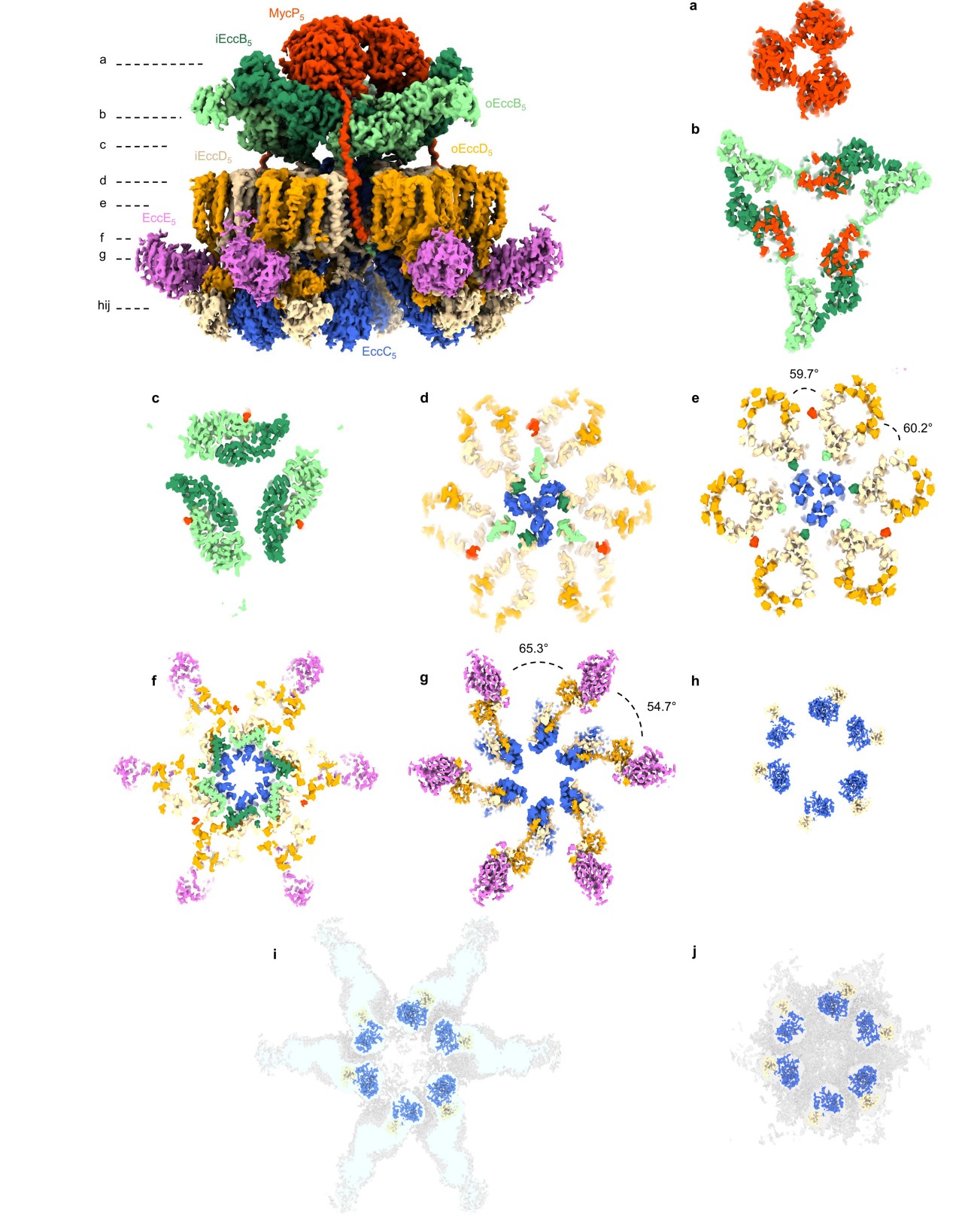

**Extended Data Fig. 5** | See next page for caption.

**Extended Data Fig. 5 | Top cross-sections through the intact ESX-5$_{mtb}$ membrane complex. a**, MycP$_5$ trimer top view, highlighting the pore formed at the periplasmic side. **b**, Section through the periplasmic assembly at the EccB$_5$–MycP$_5$ interface, showing the position of the protease domain sitting on top of inner EccB$_5$. **c**, Section through the periplasmic assembly at the EccB$_5$ dimer interface level, highlighting the MycP$_5$ linker connection to the TMH. **d**, Top view of the six membrane protomers with the closed EccC$_5$ TMH pyramid at the centre. **e**, Top cross-section through the six membrane protomers, highlighting 153 of the 165 TMHs. At the central area towards the cytosol, the three-EccC$_5$ TMH pyramid opens up in a manner similar to an iris. MycP$_5$, the protease domain of which interacts with the protomer containing inner EccB$_5$, interacts with the outer EccD$_5$ barrel of the adjacent protomer at the membrane level. At the membrane level, the angle between protomers within a dimer and between adjacent protomers of different dimers differs by only 0.5°. **f**, Top section displaying the region below the inner leaflet of the inner membrane, highlighting a further opening of the EccC$_5$ gated pore and the lower part of the EccB$_5$ basket, formed by EccB$_5$ N termini. **g**, At the cytosolic level, the angle between protomers differs to that at the membrane level. As such, the angle between protomers within a dimer grows to 65.3°, while the angle between adjacent protomers of different dimers decreases to 54.7°. The change in angles between the membrane and cytosolic regions of protomers is caused by MycP$_5$ binding, which induces a slight tilting to the protomers that it binds via inner EccD$_5$. **h**, Section through the lower region of the cytosolic bridge, containing the DUF domain of EccC$_5$ and the cytosolic domain of inner EccD$_5$. **i**, Same view as in **h**, but overlaid with the EccC$_5$ extended state, highlighting the radial extension of the EccC$_5$ NBD1, NBD2 and NBD3 almost parallel to the inner membrane. **j**, Same view as in **h**, but then overlaid with the EccC$_5$ contracted stated.

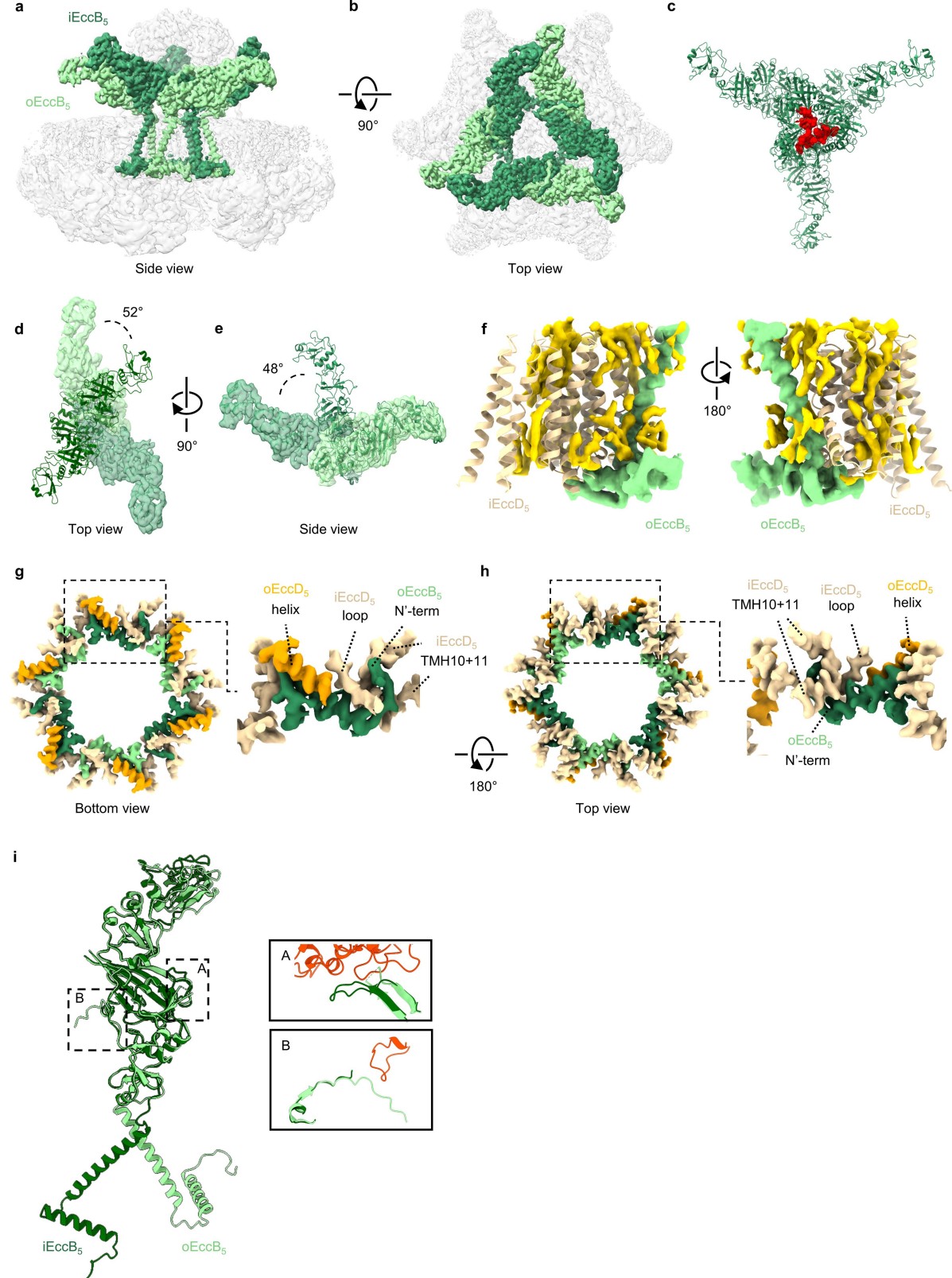

**Extended Data Fig. 6** | See next page for caption.

**Extended Data Fig. 6 | Hexameric EccB$_5$ adopts a triangular conformation in the periplasm. a**, **b**, Side (**a**) and top (**b**) view of an intact ESX-5$_{mtb}$ assembly in which inner EccB$_5$ and outer EccB$_5$ are coloured as in Fig. 1 and the rest of the components are transparent. **c**, A V-shaped EccB$_3$ dimer (PDB 6SGY) was fitted into the *M. smegmatis* ESX-3 dimer cryo-EM density (EMDB EMD-20820) together with the corresponding dimeric ESX-3 model for the membrane and cytosolic domains (PDB 6UMM) using the Chimera fit in map tool. This composite dimer model was subsequently trimerized, on the basis of our full ESX-5 map reconstructions. The clashing of EccB$_3$ periplasmic domains between the dimers, towards the central area, in this hybrid model are highlighted in red. **d**, Upon MycP$_5$ binding to the assembly, the periplasmic EccB$_5$ dimer is rotated by 52°, avoiding the clashes observed in **c**. Angles were measured by aligning the hybrid model and the ESX-5$_{mtb}$ model at the membrane level. Subsequently, centres of mass were defined for the combined R1 domains of each EccB$_3$ and EccB$_5$ dimer (at the base of the dimer) and for every R2 and R3 EccB monomer (toward the tips of the EccB dimer). Planes defined by these three points were generated for both EccB$_3$ and EccB$_5$ dimers and angles were measured between these two planes. EccB$_3$ dimer is shown as a ribbon model and EccB$_5$ model is shown as zoned density. **e**, Compared to the V-shaped EccB$_3$ dimer (ribbon model), the angle between the two EccB$_5$ monomers (zoned density) grows by 48° upon MycP$_5$ binding. EccB dimer angles were calculated by measuring the angle between the centres of mass of the R2 and R3 domains of each EccB protomer in relation to the centre of mass of both R1 domains. **f**, Side views of the TMH region of inner EccD$_5$, depicted as a ribbon model, and the TMH and N terminus of an interacting outer EccB$_5$, depicted as zoned density. An array of lipids found in the EccD$_5$ barrel (but also surrounding this inner EccD$_5$–EccB$_5$ interaction site) are depicted in gold. **g**, Bottom view of the lower cytosolic area of the EccB$_5$ basket, formed by EccB$_5$ N termini (residues 10–48) and depicted with the interacting pocket formed by TMH10, TMH11 and TMH8 (not shown for clarity) of inner EccD$_5$ of the adjacent protomer. The EccB$_5$ N terminus is also buttressed in this position by a short helix (residues 119–130) of outer EccD$_5$, which connects outer EccD$_5$ TMHs with its cytosolic domain, and also by part of the inner EccD$_5$ loop (residues 307–315) that subsequently folds along the stalk and DUF domain of EccC$_5$. **h**, Same map as in **g**, but viewed from the top. **k**, Superposition of inner EccB$_5$ and outer EccB$_5$, highlighting conformational differences between the two, which are the result of the interaction with MycP$_5$.

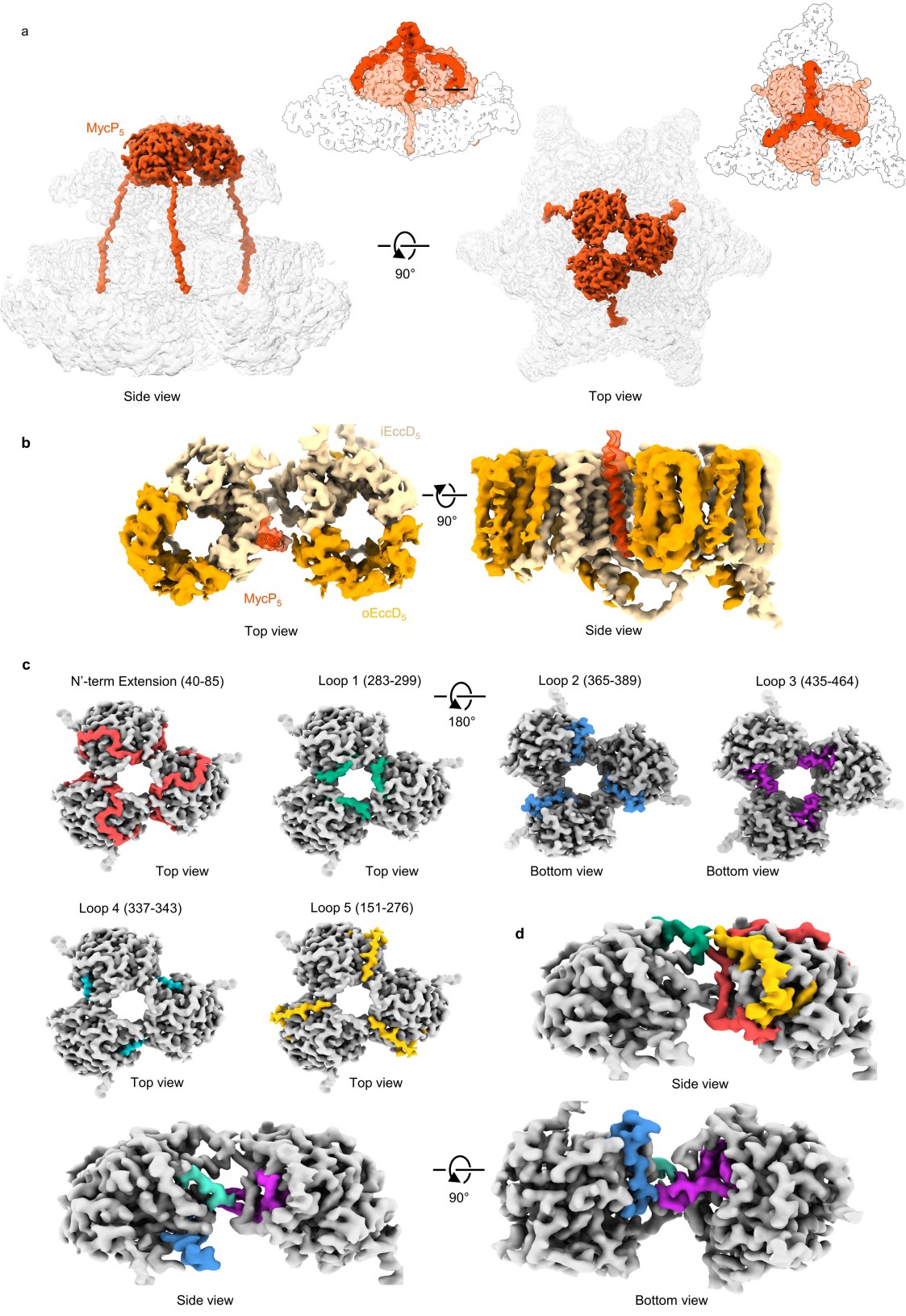

**Extended Data Fig. 7 | MycP₅ caps a periplasmic cavity with its active site directed towards the lumen. a**, Side and top view of an intact ESX-5mtb assembly with MycP₅ coloured as in Fig. 1 and the rest of the components transparent. Insets, side and top views of the periplasmic assembly with EccB₅ in white, the MycP₅ density shown in transparent red and loop 5 of MycP₅ depicted in solid red at a higher threshold, to highlight it capping the periplasmic pore. Loop 5 folds along the protease domain, towards the pore

formed by the MycP₅ trimer. At higher thresholds, loop 5 caps this pore. **b**, Top and side view of a dimer of EccD₅ barrels, of which one barrel (left) binds via inner EccD₅ to the MycP₅ TMH. **c**, Top or bottom view of MycP₅ trimers depicted in grey with the loops that are involved in MycP₅–MycP₅ interactions depicted in different colours. **d**, Side and bottom views showing the MycP₅–MycP₅ interactions mediated by the same domains depicted in the same colours as in **e**.

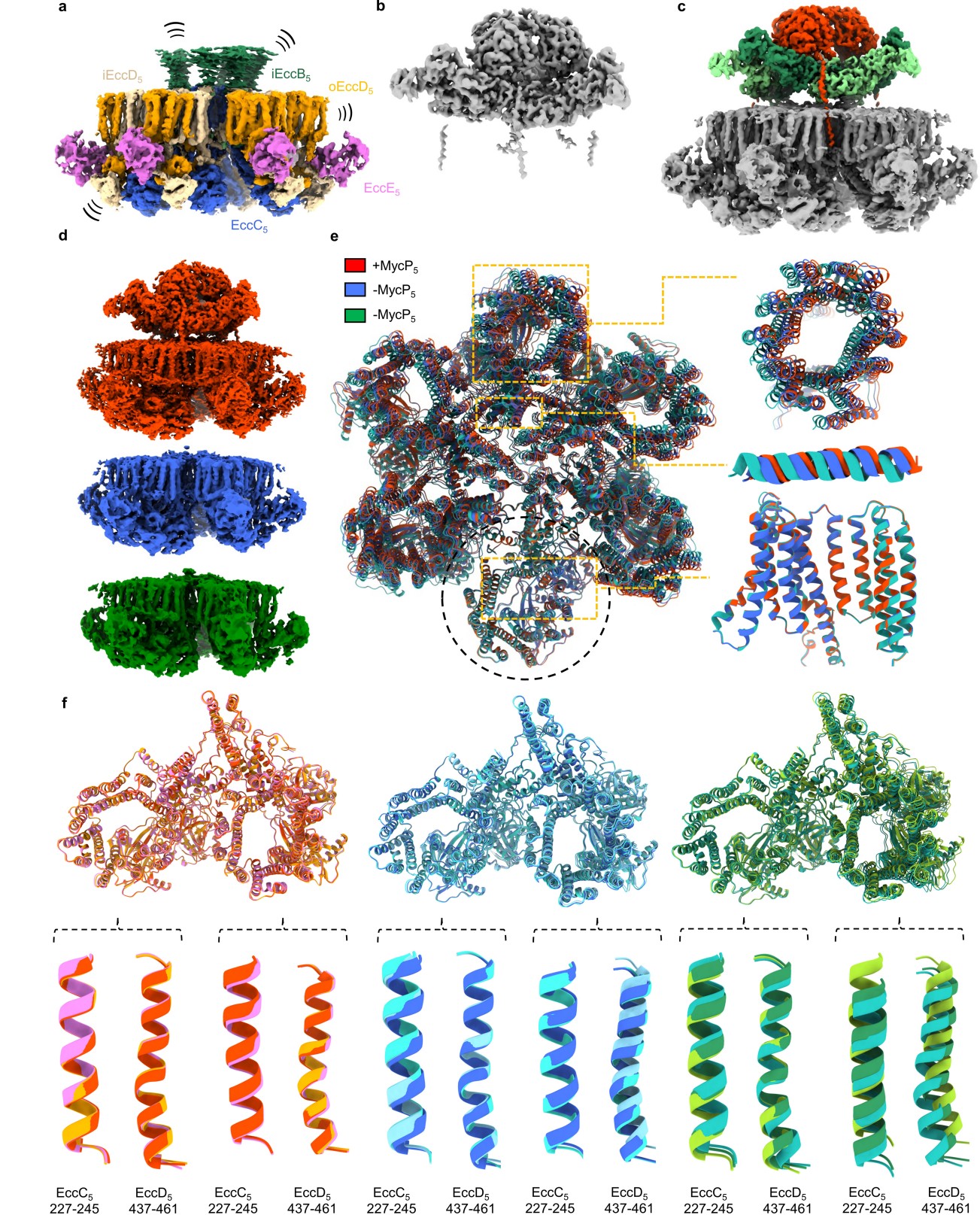

**Extended Data Fig. 8 |** See next page for caption.

**Extended Data Fig. 8 | MycP$_5$ drives hexamerization of periplasmic EccB$_5$ and complex stability. a**, Cryo-EM density map of a MycP$_5$-free ESX-5$_{mtb}$ membrane complex, zoned and coloured as in Fig. 1. In the absence of MycP$_5$, the periplasmic domains of EccB$_5$ display high flexibility. The rest of the membrane complex displays increased heterogeneity when compared to the MycP$_5$-bound map. **b**, Map of difference created by subtracting the MycP$_5$-free map from the MycP$_5$-bound map. **c**, Overlay of **a** and **b**. **d**, MycP$_5$-bound map in red and the two MycP$_5$-free maps in blue and green. **e**, A model of the MycP$_5$-bound map, in which MycP$_5$ and residues 84–504 of EccB$_5$ were removed, was fitted into the models of the two MycP$_5$-free maps, as described in Methods. Models were aligned at one EccD$_5$ barrel (dark dotted circle), revealing substantial variations and shifts between the three maps. Top inset shows that there is consistent variation between all three maps at the membrane level (EccD$_5$ barrel). Middle inset shows variations between maps at cytosolic level (EccB$_5$ N-terminal helix, residues 20–38). Bottom inset highlights inner EccD$_5$ from the EccD$_5$ barrel that was used for the alignment, showing that overall protomer structure does not change in the absence of MycP$_5$. **f**, Dimers from every individual map, colour-coded the same as in **d** (different shades), were extracted and aligned to each other on one EccD$_5$ barrel (left) as in **e**. Insets from these alignments, derived from both protomers, show that all three dimers of the MycP$_5$-bound map show little to no variation, whereas the two MycP$_5$-free maps show a higher degree of heterogeneity between dimers.

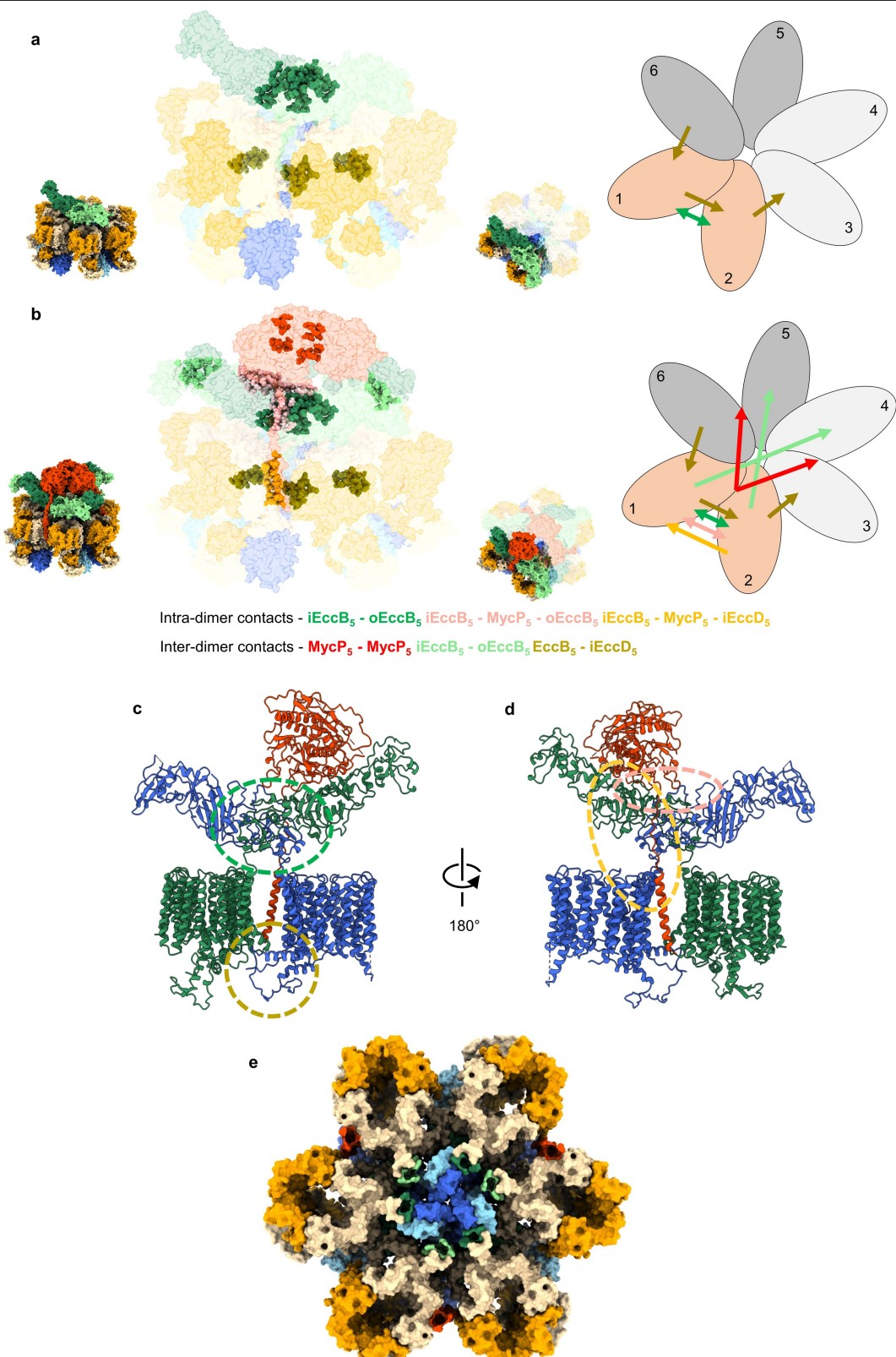

Intra-dimer contacts - **iEccB$_5$ - oEccB$_5$** **iEccB$_5$ - MycP$_5$** **oEccB$_5$ iEccB$_5$ - MycP$_5$ - iEccD$_5$**

Inter-dimer contacts - **MycP$_5$ - MycP$_5$** **iEccB$_5$ - oEccB$_5$** **EccB$_5$ - iEccD$_5$**

**Extended Data Fig. 9** | See next page for caption.

**Extended Data Fig. 9 | MycP$_5$ creates more interaction points between protomers and dimers. a**, Transparent surface model of a MycP$_5$-free map with one EccB$_5$ dimer at the periplasmic side, highlighting the interfaces of the protomers from a dimer. In the absence of MycP$_5$, the two protomers from within one dimer exhibit two interactions: an EccB$_5$–EccB$_5$ interaction between their periplasmic domains, and a cytosolic one between EccB$_5$–inner EccD$_5$. The dimer further contacts the two immediate protomers of adjacent dimers through the mentioned EccB$_5$–EccD$_5$ cytosolic interaction. **b**, Transparent surface model of the MycP$_5$-bound map, highlighting the interface of protomers from a dimer. On top of the mentioned contacts, in the presence of MycP$_5$, protomers from a dimer interact with each other at the periplasmic side through inner EccB$_5$–MycP$_5$–outer EccB$_5$, while MycP$_5$ further anchors the periplasmic assembly to the stable EccD$_5$ raft through EccB$_5$–MycP$_5$–inner EccD$_5$. MycP$_5$ also guides dimer–dimer interactions. By stabilizing the three EccB$_5$ dimers in the triangle assembly, MycP$_5$ promotes inner EccB$_5$–outer EccB$_5$ interactions between opposing protomers from adjacent dimers. Additionally, MycP$_5$ promotes dimer–dimer contacts through MycP$_5$–MycP$_5$ interactions in the periplasm. Colour-coded legend applies to **a**, **b**. **c**, **d**, Inside (**c**) and outside (**d**) view of a dimer containing EccB$_5$, MycP$_5$ and the TMHs of EccD$_5$. For purposes of clarity, one EccD$_5$–EccB$_5$ protomer is coloured in blue, and the second protomer is in green and the MycP$_5$ in red. Interactions between protomers of a dimer are highlighted and colour-coded as in **a**, **b**. **e**, Top view of a surface model missing the periplasmic domains of EccB$_5$ and MycP$_5$. The planes of protomers in which MycP$_5$ binds inner EccD$_5$ are tilted by about 5° compared to the MycP$_5$-unbound ones.

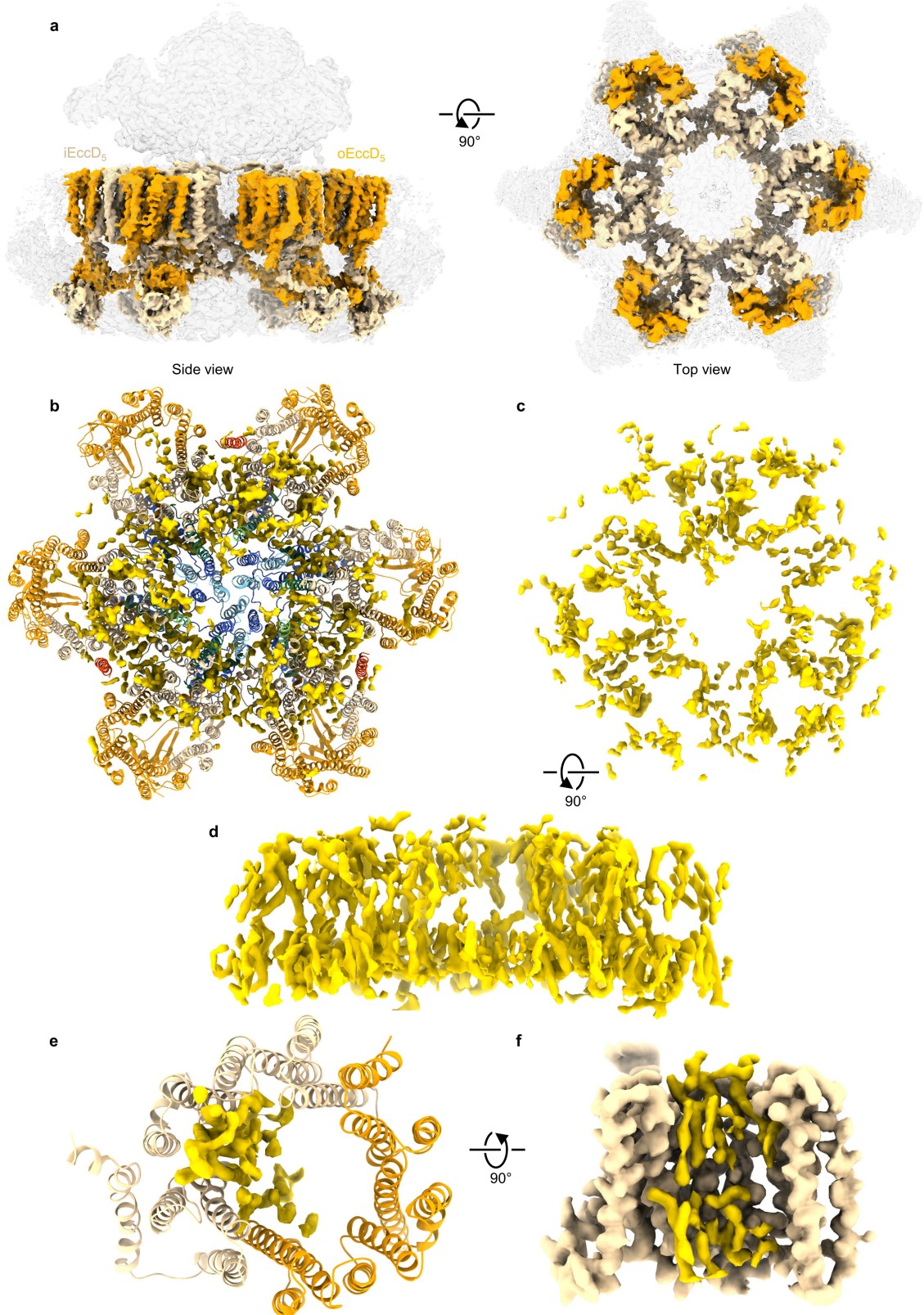

**Extended Data Fig. 10 | Six lipid-filled EccD$_5$ barrels form a central raft.**
**a**, Side and top view of an intact ESX-5$_{mtb}$ assembly, in which inner EccD$_5$ and outer EccD$_5$ are coloured as in Fig. 1 and the rest of the components are transparent. **b**, Top view of the membrane region of the ESX-5$_{mtb}$ model overlaid with observed lipids, coloured in bright yellow. **c**, Same view as **b**, but showing only the lipids. **d**, Same image as **c**, but rotated 90° to show a side view, highlighting a bilayer-like structure. **e**, Top view of an EccD$_5$ barrel with observed lipids bound to inner EccD$_5$. **f**, Side view of an inner EccD$_5$ monomer displayed as zoned density and overlaid with observed lipids.

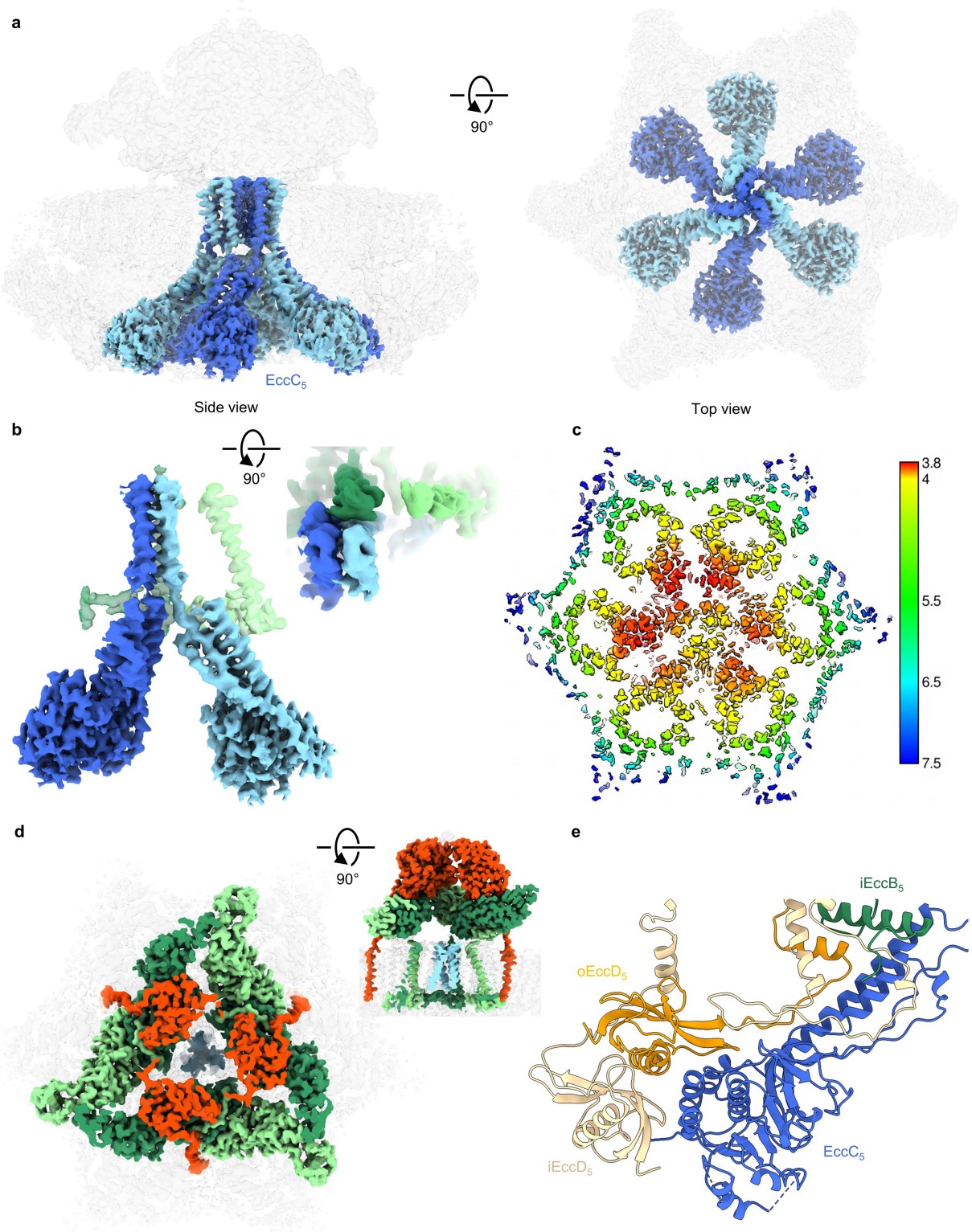

**Extended Data Fig. 11 | Three four-TMH-bundles of EccC₅ gate a central pore. a**, Side and top view of an intact ESX-5_mtb assembly, in which EccC₅ is coloured in alternating light and dark blue and the rest of the components are transparent. **b**, Extracted dimeric EccC₅ and the TMHs and N termini of EccB₅ from the same dimer. The 90° inset rotation shows that the TMH of outer EccB₅ is not contacted by the TMHs of EccC₅. **c**, Top membrane cross-section through a local-resolution map of a C1 full-complex reconstruction, displaying decreased resolution of the central space occupied by the TMHs of EccC₅, compared to the surrounding EccB₅ basket and TMHs of inner EccD₅. **d**, Top view of the full membrane complex with the EccC₅ TMH pyramid in light blue and the periplasmic EccB₅–MycP₅ in the same colours as in Fig. 1. The EccC₅ TMH pyramid aligns with the periplasmic cavity and the MycP₅-formed pore. MycP₅ top part is partially sectioned, for clarity. Inset showing a 90° rotation side cross-section of the same map. **e**, Ribbon model highlighting the structural features of the cytosolic bridge.

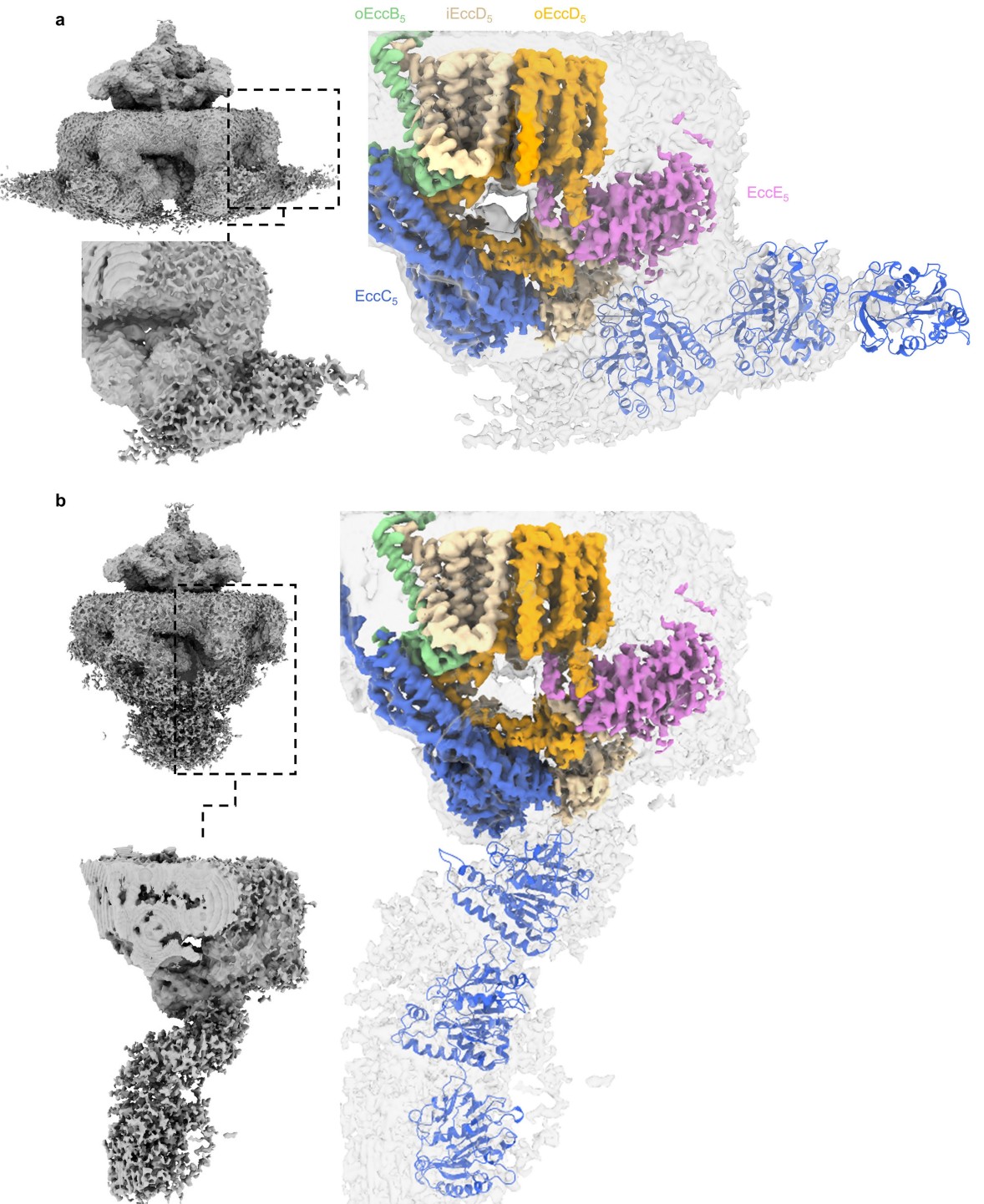

**Extended Data Fig. 12 | EccC₅ adopts two separate conformations. a**, Extended conformation in which an EccC₅ NBD1–NBD2–NBD3 model is fitted to highlight the overall position of these domains with respect to the rest of the membrane complex. **b**, As in **a**, but for the contracted conformation.

# nature research

# Reporting Summary

Nature Research wishes to improve the reproducibility of the work that we publish. This form provides structure for consistency and transparency in reporting. For further information on Nature Research policies, see our Editorial Policies and the Editorial Policy Checklist.

## Statistics

For all statistical analyses, confirm that the following items are present in the figure legend, table legend, main text, or Methods section.

| n/a | Confirmed | |
|---|---|---|
| ☐ | ☒ | The exact sample size (*n*) for each experimental group/condition, given as a discrete number and unit of measurement |
| ☒ | ☐ | A statement on whether measurements were taken from distinct samples or whether the same sample was measured repeatedly |
| ☒ | ☐ | The statistical test(s) used AND whether they are one- or two-sided *Only common tests should be described solely by name; describe more complex techniques in the Methods section.* |
| ☒ | ☐ | A description of all covariates tested |
| ☒ | ☐ | A description of any assumptions or corrections, such as tests of normality and adjustment for multiple comparisons |
| ☒ | ☐ | A full description of the statistical parameters including central tendency (e.g. means) or other basic estimates (e.g. regression coefficient) AND variation (e.g. standard deviation) or associated estimates of uncertainty (e.g. confidence intervals) |
| ☒ | ☐ | For null hypothesis testing, the test statistic (e.g. *F*, *t*, *r*) with confidence intervals, effect sizes, degrees of freedom and *P* value noted *Give P values as exact values whenever suitable.* |
| ☒ | ☐ | For Bayesian analysis, information on the choice of priors and Markov chain Monte Carlo settings |
| ☒ | ☐ | For hierarchical and complex designs, identification of the appropriate level for tests and full reporting of outcomes |
| ☒ | ☐ | Estimates of effect sizes (e.g. Cohen's *d*, Pearson's *r*), indicating how they were calculated |

*Our web collection on statistics for biologists contains articles on many of the points above.*

## Software and code

Policy information about availability of computer code

| Data collection | CryoEM: Thermo Fisher EPU V1.11 and V2.4; Negative stain EM: Thermo Fisher TIA V4.1.5; Western blotting: ChemoStar Touch (Intas Science Imaging Instruments GmbH, v. 0.5.65). |
|---|---|
| Data analysis | MotionCor2 v.1.2.1 and v.1.3, CTFfind v.4.1.13, crYOLO v.1.4, Relion 3.1-beta, Chimera v.1.13.1 and 1.14, ChimeraX v.1 and v.1.1, Pymol v.2.40, Phyre2 (unversioned), Phenix v.1.18.2, Density modification (unversioned) and EMRinger (unversioned) both within the Phenix suite, DeepEMhancer (unversioned), ISOLDE v.1.0b5, MolProbity (unversioned), Phenix.real_space_refine v.1.18-6831, PISA (unversioned). |

For manuscripts utilizing custom algorithms or software that are central to the research but not yet described in published literature, software must be made available to editors and reviewers. We strongly encourage code deposition in a community repository (e.g. GitHub). See the Nature Research guidelines for submitting code & software for further information.

## Data

Policy information about availability of data

All manuscripts must include a data availability statement. This statement should provide the following information, where applicable:

- Accession codes, unique identifiers, or web links for publicly available datasets
- A list of figures that have associated raw data
- A description of any restrictions on data availability

Cryo-EM maps have been deposited in the Electron Microscopy Database under accession codes EMD-12514 (full complex in C1), EMD-12517 (full complex in C3), EMD-12518 (periplasmic map in C1), EMD-12519 (periplasmic map in C3), EMD-12520 (cytosolic bridge), EMD-12521 (MycP5-free map 1), EMD-12522 (MycP5-free map 2), EMD-12523 (EccC5 extended state) and EMD-12525 (EccC5 contracted state). The composite model settled in the C1 and C3 full maps, periplasm in C1, cytosolic bridge, MycP5-free map 1 and MycP5-free map 2 have been deposited in the Protein Data Bank under PDB accession codes 7NP7, 7NPR, 7NPS, 7NPT, 7NPU and 7NPV respectively. All other data is available from the corresponding author upon reasonable request.

# Field-specific reporting

Please select the one below that is the best fit for your research. If you are not sure, read the appropriate sections before making your selection.

☒ Life sciences ☐ Behavioural & social sciences ☐ Ecological, evolutionary & environmental sciences

For a reference copy of the document with all sections, see nature.com/documents/nr-reporting-summary-flat.pdf

# Life sciences study design

All studies must disclose on these points even when the disclosure is negative.

| | |
|---|---|
| Sample size | Sample sizes were chosen as a maximum possible while considering practical limitations for data collection and subsequent data processing. The size of the final particle set was determined by the ability to reach resolutions better than 4 Å in 3D reconstructions. |
| Data exclusions | Data was excluded during cryo-EM data processing by removing 2D and 3D classes that did not posses high-resolution features, a standard method for cryo-EM high resolution structural determination. |
| Replication | All experiments have been successfully replicated. Solubilized membranes have been analyzed with BN-PAGE and EccB5 immunostaining three times (ED Fig. 1B). Purifications +/- nucleotides (ED Fig. 1f, g) was replicated three times. Purification without nucleotides with EM assessment has been performed three times (ED Fig. 1c, d, e) and purification with nucleotides together with cryo-EM data collection has been performed two times (ED Fig. 1 h, i, j). |
| Randomization | Particles/images were randomly partitioned for resolution and quality assesment. |
| Blinding | Blinding during data collection and analysis is not a commonly applied procedure in cryo-EM. |

# Behavioural & social sciences study design

All studies must disclose on these points even when the disclosure is negative.

| | |
|---|---|
| Study description | Not relevant to this study |
| Research sample | Not relevant to this study |
| Sampling strategy | Not relevant to this study |
| Data collection | Not relevant to this study |
| Timing | Not relevant to this study |
| Data exclusions | Not relevant to this study |
| Non-participation | Not relevant to this study |
| Randomization | Not relevant to this study |

# Ecological, evolutionary & environmental sciences study design

All studies must disclose on these points even when the disclosure is negative.

| | |
|---|---|
| Study description | Not relevant to this study |
| Research sample | Not relevant to this study |
| Sampling strategy | Not relevant to this study |
| Data collection | Not relevant to this study |
| Timing and spatial scale | Not relevant to this study |
| Data exclusions | Not relevant to this study |
| Reproducibility | Not relevant to this study |
| Randomization | Not relevant to this study |

Blinding | Not relevant to this study

Did the study involve field work? ☐ Yes ☒ No

# Reporting for specific materials, systems and methods

We require information from authors about some types of materials, experimental systems and methods used in many studies. Here, indicate whether each material, system or method listed is relevant to your study. If you are not sure if a list item applies to your research, read the appropriate section before selecting a response.

## Materials & experimental systems

| n/a | Involved in the study |
|-----|----------------------|
| ☐ | ☒ Antibodies |
| ☒ | ☐ Eukaryotic cell lines |
| ☒ | ☐ Palaeontology and archaeology |
| ☒ | ☐ Animals and other organisms |
| ☒ | ☐ Human research participants |
| ☒ | ☐ Clinical data |
| ☒ | ☐ Dual use research of concern |

## Methods

| n/a | Involved in the study |
|-----|----------------------|
| ☒ | ☐ ChIP-seq |
| ☒ | ☐ Flow cytometry |
| ☒ | ☐ MRI-based neuroimaging |

## Antibodies

Antibodies used | anti EccB5 antibody

Validation | The anti EccB5 antibody was raised against the synthetic peptide CLPMDMSPAELVVPK by Innovagen (Lund, Sweden) and has been validated in doi:10.1111/j.1365-2958.2012.08206.x. It was used as a 1:5000 dilution.

