## [Peer Review File · Nature]

Manuscript Title: Structure and dynamics of a mycobacterial type VII secretion system

Editorial Notes:

Reviewer Comments & Author Rebuttals

Reviewer Reports on the Initial Version:

Referee #1 (Remarks to the Author):

In their paper, Bunduc et al report the structure and dynamics of the ESX-5 type VII secretion system of *Mycobacterium tuberculosis*.

The authors show how the 2.32 MDa, 165 transmembrane helices-containing ESX-5 assembly is restructured and stabilized as a trimer of dimers by the MycP5 protease.

The structure of a mycobacterial ESX-5 system has been subject of a previous publication by a group of authors, who in part are also authors of the current publication. In the previous paper from 2017 they showed at 13 Å resolution the first structure of ESX-5 from *Mycobacterium xenopi*. In the current work, the authors have cloned the genes encoding the ESX-5 system of *M. tuberculosis* into a vector expressed in *M. smegmatis*, and have by this way succeeded to obtain a much more detailed structure of the ESX-5 nano-machine than in the study from 2017. The main differences between the previous structure of ESX-5 and the one in this manuscript is the difference in stoichiometry. Previously it was thought that the four core proteins of the ESX-5 complex (EccB5, EccC5, EccD5 and EccE5) assemble with equimolar stoichiometry, whereas now the authors describe the ESX-5 assembly as a trimer of dimers where each dimer is composed of two protomers of EccB5:EccC5:EccD5:EccE5 in a ratio of 1:1:2:1, and a single MycP5 copy. These data resemble the stoichiometry of a recently published structure of the ESX-3 system from *M. smegmatis*, whereby in the current work the authors have been able to co-purify the conserved MycP5 protease. This component was absent in all previously reported ESX structures despite the fact that MycP is known to be essential for Type VII secretion functioning and complex stability. Moreover, the authors report about an extended and contracted conformation of EccC5 units, which provides new insights into the putative open/closing mechanism of the system. Thus the here-presented structure of ESX-5 including MycP is a much improved structure over the previous ESX-5 structure published at 13 Å, and represents a fully-assembled structure of the ESX-5 inner membrane complex of *M. tuberculosis*. The structure allows now a much better prediction of the protein transport mechanisms compared to previous Type VII structures, and represents a platform for identification of compounds that may interfere with the function of Type VII systems, that could eventually be used as anti virulence drugs against *M. tuberculosis*.

The paper is clearly written and thus I only have minor suggestions for improvement. It would indeed be helpful if the authors could discuss in more detail how the current structure differs from previous work, and how the structure with a higher resolution helps to identify new anti-tuberculosis drugs.

Referee #2 (Remarks to the Author):

Mycobacterial type VII secretion systems (T7SS) are all composed of a core complex embedded in the inner membrane and made of the proteins EccB, EccC, EccD, EccE and MycP. In a first breakthrough, Beckham et al. revealed the molecular architecture of the ESX-5 membrane complex using negative stain electron microscopy. In particular, these authors could show that this

complex displays a 6-fold cylindrical symmetry. More recently, the cryoEM structure of the ESX-3 membrane complex was revealed by two independent studies (Famelis et al. 2019 and Poweleit et al 2019). These authors could show that the EccB, C, D and E proteins assemble as two identical promoters with 1:1:2:1 stoichiometry. However, all these CryoEM structures were lacking MycP, an essential component of the T7SS machinery and did not display the six fold symmetry that was initially described for the ESX-5 complex.

Here, Bunduc et al. describe the structure of an entire ESX-5 type VII membrane complex from *Mycobacterium tuberculosis*, including MycP. The authors reveal the details of the molecular architecture of this complex made of 6 protomers made of EccB, C, D, E and MycP with a 1:1:2:1:1 stoichiometry. They describe in great details the differences between the structures previously obtained for the ESX-3 complex and this new structure. In particular, they show that MycP is an essential structural component of the membrane complex that stabilizes the whole assembly. They also reveal that EccC cytoplasmic domains could adopt two distinct conformations. These new data provide a substantial amount of original and robust information on the T7SS assembly.

The manuscript is well written and includes beautiful figures (main and extended figures). The methods used are state of the art CryoEM. It seems that all the information embedded in the data has been properly extracted and sorted.

Minor comments:

- MycP is described as an allosteric driver of EccB hexamerisation. Allostery is an enzymology concept (regulation of an enzymatic activity by the binding of a molecule). MycP stabilizes the membrane complex assembly by forming a periplasmic dome that interacts with EccB. It is not an allosteric regulation. This should be corrected. In general the information contained in the paragraphs "three MyP5 subunits stabilize...periplasmic dome" and MycP5 acts as...complex stabilization" is slightly redundant. The main message being that MycP stabilizes the whole assembly, maybe the text could be shorter for this part. That would allow some discussion of the results (see below).

-Esx-1 T7SS is the major virulence factor for *M. Tuberculosis*. Are the data obtained on ESX-5 T7SS valid for ESX-1. Are the components conserved? Is it likely that this magnificent architecture will be conserved? Or is it specific to ESX-5? Why is so difficult to obtain the structure of the ESX-1 T7SS. please comment in the conclusion of the MS (if space allows). This would give more perspective to this important results.

Author Rebuttals to Initial Comments:

Response

We would like to thank all the reviewers for carefully reading our manuscript and providing their expert opinion. We are grateful for the constructive feedback which helped us to further improve the manuscript. We have addressed all of the referees' points below and have updated the text and figures in the revised manuscript accordingly.

Once again, we would like to thank all reviewers for the critical evaluation of our data, and we are looking forward to share our discoveries with the broader scientific community and are grateful that the reviewers recommend publication in this journal.

Response to Comments from Reviewer 1

Comment 1:

It would indeed be helpful if the authors could discuss in more detail how the current structure differs from previous work, and how the structure with a higher resolution helps to identify new anti-tuberculosis drugs.

Response:

We thank the reviewer for his/her comment. We have amended the manuscript to further highlight structural differences and similarities between our work and previous publications, while taking into consideration the size restraints. We show a “side by side” comparison between the EccB₃ and EccB₅ periplasmic assemblies (Page 4, lines 82-85; Extended Data Figure 6 b, c, d, e), discuss the similar structural features at protomer level, between ESX-3_{msm} and ESX-5_{mtb} (page 3, lines 65-72; page 6, lines 147-148) and also make a short parallel with the previous hexameric, low resolution, MycP₅ free map of ESX-5 from *M. xenopi* (page 5, lines 139-141).

Next to these major structural differences, we would also like to point out that our work deals with an intact T7SS inner membrane complex which comes from the human pathogen *M. tuberculosis* itself, and therefore this structure can be used to guide drug design studies. This is also now specified in the text (page 7, lines 208-210).

Response to Comments from Reviewer 2

Comment 1:

MycP is described as an allosteric driver of EccB hexamerisation. Allostery is an enzymology concept (regulation of an enzymatic activity by the binding of a molecule). MycP stabilizes the membrane complex assembly by forming a periplasmic dome that interacts with EccB. It is not an allosteric regulation.

Response:

We agree with the reviewer's comment and have adapted the text accordingly in the revised manuscript.

Comment 2:

In general, the information contained in the paragraphs “three MyP₅ subunits stabilize...periplasmic dome” and MycP₅ acts as...complex stabilization” is slightly redundant. The text could be shorter for this part to allow some discussion of the results.

Response:

We thank the reviewer for his comment. We have shortened the text overall and have added more discussion points throughout the manuscript, considering the size restraints (page 5, lines 112-116 and 134-141; page 6, lines 177-181; page 6, lines 177-181; page 7, lines 201-210).

Comment 3:

ESX-1 T7SS is the major virulence factor for M. tuberculosis. Is the data obtained on ESX-5 T7SS valid for ESX-1? Are the components conserved? Is it likely that this magnificent architecture will be conserved? Or is it specific to ESX-5? Why is it so difficult to obtain the structure of the ESX-1 T7SS. This would give more perspective to these important results.

Response:

Considering the high sequence conservation between the five membrane components of T7SSs (highlighted also in our Supplementary Fig. 2&3) and the structural similarities to the ESX-3 dimeric subunit or other individual components, we believe that our intact structure constitutes a general blueprint for the assembly of other T7SSs, including the important ESX-1 system. We thank the reviewer for his comment and have added this to our manuscript.

We did not speculate on the difficulty of obtaining a structure of the ESX-1 system, as we have no direct experience with this complex, but do believe that our current methodology of expression and purification could be helpful.